# SEQUENTIAL GRADIENT CODING FOR STRAGGLER MITIGATION

**M. Nikhil Krishnan** *
Indian Institute of Technology Palakkad
nikhilkrishnan.m@gmail.com

**M. Reza Ebrahimi** *
University of Toronto
mr.ebrahimi@mail.utoronto.ca

**Ashish Khisti**
University of Toronto
akhisti@ece.utoronto.ca

## ABSTRACT

In distributed computing, slower nodes (stragglers) usually become a bottleneck. Gradient Coding (GC), introduced by Tandon et al., is an efficient technique that uses principles of error-correcting codes to distribute gradient computation in the presence of stragglers. In this paper, we consider the distributed computation of a sequence of gradients $\{g(1), g(2), \ldots, g(J)\}$, where processing of each gradient $g(t)$ starts in round-$t$ and finishes by round-$(t + T)$. Here $T \geq 0$ denotes a delay parameter. For the GC scheme, coding is only across computing nodes and this results in a solution where $T = 0$. On the other hand, having $T > 0$ allows for designing schemes which exploit the temporal dimension as well. In this work, we propose two schemes that demonstrate improved performance compared to GC. Our first scheme combines GC with selective repetition of previously unfinished tasks and achieves improved straggler mitigation. In our second scheme, which constitutes our main contribution, we apply GC to a subset of the tasks and repetition for the remainder of the tasks. We then multiplex these two classes of tasks across workers and rounds in an adaptive manner, based on past straggler patterns. Using theoretical analysis, we demonstrate that our second scheme achieves significant reduction in the computational load. In our experiments, we study a practical setting of concurrently training multiple neural networks over an AWS Lambda cluster involving 256 worker nodes, where our framework naturally applies. We demonstrate that the latter scheme can yield a 16% improvement in runtime over the baseline GC scheme, in the presence of naturally occurring, non-simulated stragglers.

## 1 INTRODUCTION

We consider a distributed system consisting of a master and $n$ computing nodes (will be referred to as workers). We are interested in computing a sequence of gradients $g(1), g(2), \ldots, g(J)$. Assume for simplicity that there are no dependencies between the gradients (we will relax this assumption later in Sec. 2). If we naively distribute the computation of each gradient among $n$ workers, delayed arrival of results from any of the workers will become a bottleneck. Such a delay could be due to various reasons such as slower processing at workers, network issues leading to communication delays etc. Irrespective of the actual reason, we will refer to a worker providing delayed responses to the master, as a straggler. In a recent work Tandon et al. (2017), the authors propose Gradient Coding (GC) to distribute computation of a single gradient across multiple workers in a straggler-resilient manner. Using $(n, s)$-GC, the master is able to compute the gradient as soon as $(n - s)$ workers respond ($s$ is an integer such that $0 \leq s < n$). Adapting GC to our setting, we will have a scheme where $g(t)$ is computed in round-$t$ and in every round, up to $s$ stragglers are tolerated (the concept of round will be formally introduced in Sec. 2). Experimental results (Yang et al., 2019) have demonstrated that intervals of "bad" rounds (where each round has a large number of stragglers) are followed by "good" rounds (where there are relatively fewer number of stragglers), leading to a natural temporal diversity.

---

*Equal contribution authors.

Note that GC will enforce a large $s$, as dictated by the number of stragglers expected in bad rounds, leading to a large computational load per worker. The natural question to ask here is that: *do there exist better coding schemes that exploit the temporal diversity and achieve better performance?*. In this paper, we present *sequential gradient coding schemes* which answer this in the affirmative.

## 1.1 SUMMARY OF CONTRIBUTIONS

In contrast to existing GC approaches where coding is performed only across workers, we propose two coding schemes which explore coding across rounds as well as workers.

- Our first scheme, namely, Selective-Reattempt-Sequential Gradient Coding (SR-SGC) (Sec. 3.2), is a natural extension of the $(n, s)$-GC scheme, where unfinished tasks are selectively reattempted in future rounds. Despite the simplicity, SR-SGC scheme tolerates a strict superset of straggler patterns compared to $(n, s)$-GC, for the same computational load.

- In our more involved second scheme, namely, Multiplexed-Sequential Gradient Coding (M-SGC) scheme (Sec. 3.3), we divide tasks into two sets; those which are protected against stragglers via reattempts and those which are protected via GC. We then carefully multiplex these tasks to obtain a scheme where computational load per worker is significantly reduced. In particular, the load decreases by a factor of $s$ when compared to the $(n, s)$-GC scheme and is close to the information theoretic limit in certain conditions.

- Our experiments (Sec. 4) on an AWS Lambda cluster involving 256 worker nodes show that the M-SGC scheme achieves a significant reduction of runtime over GC, in real-world conditions involving naturally occurring, non-simulated stragglers.

## 1.2 RELATED WORK

We provide a brief overview on the use of erasure codes for straggler mitigation in Appendix B. The terminology of "sequential gradient coding" initially appears in Krishnan et al. (2021). Both the current paper and the work Krishnan et al. (2021) extend the classical GC setting (Tandon et al., 2017), by exploiting the temporal dimension. The authors of Krishnan et al. (2021) provide a *non-explicit* coding scheme that is resilient against communication delays and does not extend when stragglers are due to computational slowdowns. In contrast, we propose two *explicit* coding schemes that are oblivious of the reason for straggling behavior. Moreover, under equivalent straggler conditions, the computational load requirements of the newly proposed SR-SGC scheme and the scheme in Krishnan et al. (2021) are identical, whereas the new M-SGC scheme offers a significantly smaller load requirement. The works Ye & Abbe (2018); Kadhe et al. (2020) explore a trade-off between communication and straggler resiliency in GC. Several variations to the classical GC setting appear in papers Raviv et al. (2020); Halbawi et al. (2018); Wang et al. (2019a;b); Maity et al. (2019).

There is a rich literature on distributing matrix multiplication or more general, polynomial function computation (see Yang et al. (2019); Yu et al. (2017); Lee et al. (2018); Ramamoorthy et al. (2019); Subramaniam et al. (2019); Yu et al. (2019); Dutta et al. (2020); Yu et al. (2020); Ramamoorthy et al. (2020) and references therein). In Krishnan et al. (2020), the authors introduce a sequential matrix multiplication framework, where coding across temporal dimension is exploited to reduce the cumulative runtime for a sequence of matrix multiplication jobs. While we also explore the idea of introducing temporal dimension to obtain improved coding schemes, extending the approach in Krishnan et al. (2020) to our setting yields an inferior solution requiring a large computational load.

## 2 SEQUENTIAL GRADIENT CODING SETTING

For integers $a, b$, let $[a : b] \triangleq \{i \mid a \leq i \leq b\}$ and $[a : b]^* \triangleq \{i \mod n \mid a \leq i \leq b\}$. For integer $c$, we have $c + [a : b]^* \triangleq \{c + i \mid i \in [a : b]^*\}$. Workers are indexed by $[0 : n - 1]$. We are interested in computing a sequence of $J$ gradients $\{g(i)\}_{i \in [1:J]}$. Computation of gradient $g(i)$ is referred to as *job-i*. All gradients are computed with respect to a single data set $\mathbf{D}$ (this assumption is just for simplicity in exposition and our schemes can easily be adapted to include multiple data sets).

*Data placement*: Master partitions $\mathbf{D}$ into $\eta$ data chunks $\{\mathbf{D}_0, \mathbf{D}_1, \ldots, \mathbf{D}_{\eta-1}\}$, possibly of different sizes. Each worker-$i$ stores data chunks $\{\mathbf{D}_j\}_{j \in D_i}$, where $D_i \subseteq [0 : \eta - 1]$. Let $g_j(t)$ denote the $t$-th

partial gradient computed with respect to $\mathbf{D}_j$, i.e., $\sum_{j \in [0:\eta-1]} g_j(t) = g(t)$. Naturally, we also say that the partial gradient $g_j(t)$ *corresponds to* job-$t$.

*Encoding*: Master operates based on a certain concept of *rounds* and it takes $J + T$ rounds to finish computing all the $J$ gradients. Here, $T \geq 0$ is a system parameter which takes integer values. Let $t \in [1 : J + T]$. In round-$t$, worker-$i$ computes $\gamma_i(t)$ partial gradients with respect to a subset of data chunks it stores. These partial gradients may correspond to any of the jobs $[1 : t]$. In other words, the partial gradients computed by worker-$i$ in round-$t$ are from within the set $\{g_j(t')\}_{t' \in [1:t], j \in D_i}$. The process of worker-$i$ computing $\gamma_i(t)$ partial gradients will be referred to as a *task*. Once the $\gamma_i(t)$ partial gradients are computed in round-$t$, worker-$i$ returns a *task result* $\phi_i(t)$, which is a function of the $\gamma_i(t)$ partial gradients, via an *encoding* step.

*Identification of stragglers*: In each round-$t$, let $\kappa(t)$ denote the time (in seconds) taken for the fastest worker (say, worker-$i$) to return the task result to the master. Master then waits for $\mu\kappa(t)$ ($\mu > 0$ is a *tolerance parameter*) more seconds and possibly more workers will return their task results during this time. Any worker which does not return its task result during this time will be marked as a straggler. Any pending tasks of stragglers will then be canceled and the system will move to the next round. Clearly, if worker-$i$ is a straggler in round-$t$, the task undertaken by it has "failed" and task result $\phi_i(t)$ will not be returned to the master. We note that determining stragglers using the $\mu$ parameter is in line with the earlier work Krishnan et al. (2020).

*Decoding*: In the end of round-$(t + T)$, master attempts to produce (decode) $g(t)$ using all available results from the set $\{\phi_i(t')\}_{i \in [0:n-1], t' \in [1:t+T]}$. i.e., master tries to finish job-$t$, in the end of round-$(t + T)$. Decoding is assumed to incur negligible computational cost, in comparison to that for partial gradient computation. This assumption can easily be justified for our proposed schemes, as they require only finite memory. As workers potentially start computing partial gradients corresponding to any job-$t$ only from round-$t$ onwards, the parameter $T$ may be regarded as *delay* in finishing a job.

*Normalized computational load per worker*: Let $d$ denote the number of data points in $\mathbf{D}$. Let $d_i(t)$ denote the total number of data points across which worker-$i$ computes $\gamma_i(t)$ partial gradients in round-$t$. The normalized (computational) load at worker-$i$ in round-$t$ is then given by: $L_i(t) \triangleq \frac{d_i(t)}{d}$.

**Remark 2.1.** Any dependencies existing between jobs need to be managed by choosing the $T$ parameter accordingly. Suppose job-$i_2$ is dependent on the computation of $g(i_1)$, where $i_1 < i_2$. One may choose here a $T$ satisfying $T \leq i_2 - i_1 - 1$. In our experiments presented in Sec. 4, we concurrently train $M$ neural network models, where jobs $Mi + 1, \ldots, M(i + 1)$ are $i$-th iterations ($i = 0, 1, \ldots$) in the training of models $1, \ldots, M$, respectively. Here, we need $T \leq M - 1$.

**Remark 2.2.** The flexibility of $T > 0$ (coding across rounds) enables us to design coding schemes which feature lower computational load and tolerate practically motivated straggler models. This in turn leads to smaller cumulative runtime (in seconds) for finishing $J$ jobs (in $J + T$ rounds).

## 2.1   STRAGGLER MODELS

For the sake of our analysis and code design, we will be referring to the following three straggler models. However, as explained in Remark 2.3 at the end of this section, and validated in Sec. 4, our coding schemes apply to any naturally occurring straggler patterns.

$(B, W, \lambda)$-*bursty straggler model*: Based on experiments over Amazon EC2 cluster, the authors of Yang et al. (2019) observe that a 2-state Gilbert-Elliot (GE) model (see Appendix C for more details) can be used to track transitions of workers from being stragglers to non-stragglers and vice-versa. However, while designing codes for straggler mitigation, a deterministic counterpart for the GE model is more ideal. Sliding-window-based deterministic models have been employed in many works as a reliable approximation of GE model; for instance, in the early classical work Forney (1971), in Saberi et al. (2019) in the context of control systems, Martinian & Sundberg (2004) in the context of low-latency communications and Krishnan et al. (2020) in the context of distributed matrix multiplication. We will now define the $(B, W, \lambda)$-bursty straggler model. Let $S_i(t)$ be an indicator function which is 1 if worker-$i$ is a straggler in round-$t$ and 0 otherwise. The straggler model is defined by the following two properties:

1. (Spatial correlation): In every window $W_j$ of the form $[j : j + W - 1]$ consisting of $W$ consecutive rounds starting at round-$j$, there are at most $\lambda \leq n$ distinct stragglers. i.e., size of the set of workers given by $\{i \mid S_i(t) = 1 \text{ for some } t \in W_j\}$ is at most $\lambda$.

2. (Temporal correlation): For any worker-$i$, first and last straggling slots (if any) are not more than $B-1$ rounds apart in every window $W_j$. i.e., if $S_i(t) = 1$, for some $t \in W_j$, then $S_i(l) = 0$ for all $l \in [t + B : j + W - 1]$.

Clearly, the parameters $\lambda, B$ satisfy: $0 \le \lambda \le n, 1 \le B \le W$.

$(N, W', \lambda')$-*arbitrary straggler model*: This is a natural extension of the $(B, W, \lambda)$-bursty straggler model where we consider arbitrary stragglers instead of bursty stragglers. As in the bursty model, in every window $W'_j$ of the form $[j : j + W' - 1]$, there are at most $\lambda'$ distinct stragglers. Moreover, for every worker-$i$, $\sum_{t \in W'_j} S_i(t) \le N$. The parameters $\lambda', N$ satisfy: $0 \le \lambda' \le n, 0 \le N \le W'$.

$s$-*stragglers-per-round model*: In this model, in each round, at most $s$ workers can be stragglers. Here $s$ is an integer satisfying $0 \le s < n$.

**Remark 2.3.** Coding schemes that we present in the paper (see Sec. 3.2 and 3.3) are designed to tolerate straggler patterns conforming to a "mixture" of these deterministic straggler models. As these models are a design choice, the ground truth associated with the actual straggler behavior need not always conform to them. For instance, we evaluate the performance of our coding schemes on a large AWS cluster (see Sec. 4), in the presence of naturally occurring stragglers. Clearly, the actual straggler behavior here need not be conforming to the straggler model used during code design. However, even then, every job-$t$ will still be finished by the end of round-$(t + T)$. We ensure this in the following manner. In any round, if the actual straggler pattern deviates from the straggler model assumption, the master will wait for stragglers to return results in that round. This way, the actual straggler pattern will "effectively" continue conform to the assumed straggler model.

# 3 SEQUENTIAL GRADIENT CODING SCHEMES

## 3.1 PRELIMINARIES: GRADIENT CODING

We present here a summary of the $(n, s)$-GC scheme. The data set $\mathbf{D}$ is partitioned into $\eta \triangleq n$ equally sized data chunks $\mathbf{D}_0, \mathbf{D}_1, \ldots, \mathbf{D}_{n-1}$. Worker-$i$ stores $s + 1$ data chunks $\{\mathbf{D}_j\}_{j \in [i:i+s]^*}$. With respect to each $\mathbf{D}_j$ stored, worker-$i$ computes the partial gradient $g_j$. Hence, worker-$i$ computes the following $s + 1$ partial gradients: $\{g_j\}_{j \in [i:i+s]^*}$. Worker-$i$ then transmits the result, which is a linear combination of the $s + 1$ partial gradients, $\ell_i = \sum_{j \in [i:i+s]^*} \alpha_{i,j} g_j$. The coefficients $\{\alpha_{i,j}\}$ are designed so that if master has access to results returned by any $n - s$ out of $n$ workers, $g \triangleq g_0 + g_1 + \cdots + g_{n-1}$ can be computed. In other words, for any $\mathcal{W} \subseteq [0 : n-1]$ with $|\mathcal{W}| = n-s$, there exist coefficients $\{\beta_{\mathcal{W},w}\}_{w \in \mathcal{W}}$ such that $g = \sum_{w \in \mathcal{W}} \beta_{\mathcal{W},w} \ell_w$. Clearly, the scheme tolerates $s$ stragglers. Applying the $(n, s)$-GC scheme to our framework, we get a scheme which computes $g(t)$ in round-$t$ (i.e., delay $T = 0$). Each worker-$i$ in round-$t$ works on a task corresponding to job-$t$. Specifically, worker-$i$ computes $\{g_j(t)\}_{j \in [i:i+s]^*}$ and returns $\ell_i(t) = \sum_{j \in [i:i+s]^*} \alpha_{i,j} g_j(t)$. Normalized load is given by $L_{\text{GC}} = \frac{s+1}{n}$. This scheme clearly tolerates any straggler pattern conforming to the $s$-stragglers-per-round model. We will now present both our proposed schemes.

## 3.2 SELECTIVE-REATTEMPT-SEQUENTIAL GRADIENT CODING (SR-SGC) SCHEME

SR-SGC scheme is a natural extension of the $(n, s)$-GC scheme. We begin with $(n, s)$-GC as the "base scheme" and whenever there are more stragglers than the base scheme can handle, we will carefully reattempt certain tasks across time. This simple idea helps the scheme tolerate a strict superset of straggler patterns compared to the classical $(n, s)$-GC scheme for the same normalized load (we present the formal statement in Prop. 3.1).

*Design parameters*: The parameter set is given by $\{n, B, W, \lambda\}$, where $0 < \lambda \le n, B > 0$ and $B$ divides $(W - 1)$. i.e., there exists an integer $x \ge 1$ such that $W = xB + 1$. We set $s \triangleq \lceil \frac{B\lambda}{W-1+B} \rceil = \lceil \frac{\lambda}{x+1} \rceil$. The scheme incurs a delay of $T = B$ and normalized load $L_{\text{SR-SGC}} = \frac{s+1}{n}$.

*Scheme outline*: Recall the notation $\ell_i(t)$ presented in Sec. 3.1. In round-$t$, worker-$i$ will attempt to compute either $\ell_i(t)$ or else, $\ell_i(t - B)$ as we will see now. Using the property of $(n, s)$-GC, a job can be finished if master receives $(n - s)$ task results corresponding to that job. In round-$t$, if master finds out that $(n - \nu) < (n - s)$ task results corresponding to job-$(t - B)$ are received in round-$(t - B)$, the minimum additionally required number of $(\nu - s)$ tasks corresponding to job-$(t - B)$ will be

attempted in round-$t$. These tasks will be attempted by workers who did not previously return task results corresponding to job-$(t - B)$ in round-$(t - B)$. Rest of the $(n - \nu + s)$ workers will attempt tasks corresponding to job-$t$. On the other hand, if job-$(t - B)$ is already finished in round-$(t - B)$, all tasks in round-$t$ correspond to job-$t$. Let $\mathcal{N}(t)$ denote the number of task results corresponding to job-$t$ returned to master in round-$t$. In Algorithm 1, we formally describe the exact task assignments. As jobs are indexed in the range $[1 : J]$, for consistency in notation, we assume that all task results corresponding to job-$t'$ are known to master by default (i.e., $\mathcal{N}(t') = n$), whenever $t' \notin [1 : J]$.

---

**Algorithm 1** Algorithm used by master to assign tasks in round-$t$

---

Initialize $\delta \triangleq \mathcal{N}(t - B)$.
**for** $i \in [0 : n - 1]$ **do**
    **if** $\delta < n - s$ **and** $\ell_i(t - B)$ is not returned previously by worker-$i$ in round-$(t - B)$ **then**
        Worker-$i$ attempts to compute $\ell_i(t - B)$.        $\triangleright$ Recall the definition of $\ell_i(.)$ in Sec. 3.1
        Set $\delta = \delta + 1$
    **else**
        Worker-$i$ attempts to compute $\ell_i(t)$.

---

**Proposition 3.1.** *Consider the SR-SGC scheme designed for parameters $\{n, B, W, \lambda\}$, where $0 < \lambda \le n, B > 0, W = xB + 1, T = B, s \triangleq \lceil \frac{B\lambda}{W - 1 + B} \rceil$ and $L_{SR\text{-}SGC} = \frac{s+1}{n}$. The scheme then tolerates any straggler pattern which conforms to either (i) the $(B, W, \lambda)$-bursty straggler model or else, (ii) the $s$-stragglers-per-round model, when restricted to any window of $W$ consecutive rounds.*

The proof of Prop. 3.1 requires a careful analysis of dependencies between successive rounds due to repetition of tasks. We defer the proof to Appendix D.

**Remark 3.1.** Let $\lambda < n$. Without selective repetition, classical GC requires $s = \lambda$ in order to tolerate the $(B, W, \lambda)$-bursty straggler model. In SR-SGC, we are able to choose a lower $s$ value of $\lceil \frac{B\lambda}{W - 1 + B} \rceil$ and as a result, normalized load is lower as well. In another perspective, SR-SGC scheme tolerates a superset of straggler patterns compared to the GC scheme for the same normalized load.

### 3.3 MULTIPLEXED-SEQUENTIAL GRADIENT CODING (M-SGC) SCHEME

In contrast to the SR-SGC scheme (where all tasks are coded using a base $(n, s)$-GC scheme), a large fraction of computations in the M-SGC scheme are uncoded (thus incurs no computational overheads). This results in the M-SGC scheme achieving a significantly lower normalized load compared to SR-SGC or GC schemes. Such a lower normalized load eventually translates to reduced cumulative runtimes as we see in Sec. 4.

The parameter set for M-SGC scheme is given by $\{n, B, W, \lambda\}$, where $0 \le \lambda < n, 0 < B < W$. The scheme incurs a delay of $T = W - 2 + B$. A minor modification of the scheme to cover the $\lambda = n$ scenario is discussed in Remark 3.2. As M-SGC scheme is more involved compared to the SR-SGC scheme, we initially present an example which highlights the ideas involved in M-SGC scheme. Subsequently, we provide the general coding scheme in Sec. 3.3.2. The scheme tolerates straggler patterns which conform to either $(B, W, \lambda)$-bursty straggler model or $(N = B, W' = W + B - 1, \lambda' = \lambda)$-arbitrary straggler model (see Prop. 3.2).

### 3.3.1 EXAMPLE

*Data placement*: Consider parameters $\{n = 4, B = 2, W = 3, \lambda = 2\}$. Assume that data set $\mathbf{D}$ contains $d$ data points. $\mathbf{D}$ is partitioned into 16 data chunks of unequal sizes $\{\mathbf{D}_0, \ldots, \mathbf{D}_{15}\}$. Data chunks $\mathcal{D}_1 \triangleq \{\mathbf{D}_0, \ldots, \mathbf{D}_7\}$ are of equal size and contain $\frac{3}{32}d$ data points each. Similarly, data chunks $\mathcal{D}_2 \triangleq \{\mathbf{D}_8, \ldots, \mathbf{D}_{15}\}$ contain $\frac{1}{32}d$ data points each. These 16 data chunks are distributed across workers in the following manner; worker-0: $\{\mathbf{D}_0, \mathbf{D}_1, \mathbf{D}_8, \mathbf{D}_9, \mathbf{D}_{10}, \mathbf{D}_{12}, \mathbf{D}_{13}, \mathbf{D}_{14}\}$, worker-1: $\{\mathbf{D}_2, \mathbf{D}_3, \mathbf{D}_9, \mathbf{D}_{10}, \mathbf{D}_{11}, \mathbf{D}_{13}, \mathbf{D}_{14}, \mathbf{D}_{15}\}$, worker-2: $\{\mathbf{D}_4, \mathbf{D}_5, \mathbf{D}_{10}, \mathbf{D}_{11}, \mathbf{D}_8, \mathbf{D}_{14}, \mathbf{D}_{15}, \mathbf{D}_{12}\}$ and worker-3: $\{\mathbf{D}_6, \mathbf{D}_7, \mathbf{D}_{11}, \mathbf{D}_8, \mathbf{D}_9, \mathbf{D}_{15}, \mathbf{D}_{12}, \mathbf{D}_{13}\}$. Note that data chunks in $\mathcal{D}_1$ are not replicated, whereas each data chunk in $\mathcal{D}_2$ is replicated 3 times. Each job-$t$ is finished when master computes $g(t) = g_0(t) + \cdots + g_{15}(t)$. A high level idea of the coding scheme is as follows. Failed partial gradient computations with respect to data chunks $\mathcal{D}_1$ will be reattempted across rounds. Partial gradient computations with respect to data chunks $\mathcal{D}_2$ will be made straggler-resilient by employing the $(4, 2)$-GC scheme.

*Diagonally interleaved mini-tasks:* The task performed by each worker-$i$ in round-$t$ consists of $W - 1 + B = 4$ sequentially performed *mini-tasks* $\mathcal{T}_i(t; 0), \ldots, \mathcal{T}_i(t; 3)$. If worker-$i$ is a straggler in round-$t$, results of mini-tasks $\mathcal{T}_i(t; 0), \ldots, \mathcal{T}_i(t; 3)$ will not reach master and as far as master is concerned, all of them "failed". Conversely, if worker-$i$ is a non-straggler in round-$t$, all the four mini-task results will reach master in the end of round-$t$. Mini-tasks $\mathcal{T}_i(t; 0), \mathcal{T}_i(t + 1; 1), \ldots, \mathcal{T}_i(t + 3; 3)$ involve partial gradient computations corresponding to job-$t$ (see Fig. 5 in the Appendix).

*Fixed mini-task assignment*: Computations done as part of first two mini-tasks $\mathcal{T}_i(t; 0), \mathcal{T}_i(t; 1)$ are fixed for all $t$. Specifically, mini-tasks along the "diagonal" $\mathcal{T}_i(t; 0)$ and $\mathcal{T}_i(t+1; 1)$ involve computing $g_{2i}(t)$ and $g_{2i+1}(t)$, respectively.

*Adaptive mini-task assignment*: The other two mini-tasks $\mathcal{T}_i(t + 2; 2), \mathcal{T}_i(t + 3; 3)$, which also involve partial gradient computations corresponding to job-$t$, are assigned adaptively, based on the straggler patterns seen in previous rounds. Specifically, if master did not receive $g_{2i}(t)$ in round-$t$, $\mathcal{T}_i(t + 2; 2)$ involves computing $g_{2i}(t)$. Else if master did not receive $g_{2i+1}(t)$ in round-$(t + 1)$, $\mathcal{T}_i(t + 2; 2)$ involves computing $g_{2i+1}(t)$. In a similar manner, if master did not receive $g_{2i+1}(t)$ in rounds $(t + 1)$ and $(t + 2)$, $\mathcal{T}_i(t + 3; 3)$ involves computing $g_{2i+1}(t)$. Note that so far we have described only partial gradient computations with respect to data chunks in $\mathcal{D}_1$. In round-$(t + 2)$, if $g_{2i}(t)$ and $g_{2i+1}(t)$ are already available to master, $\mathcal{T}_i(t + 2, 2)$ will involve computation of three partial gradients $\{g_l(t)\}_{l \in 8 + [i:i+2]^*}$ and obtaining the linear combination $\ell_{i,0}(t) = \sum_{j \in [i:i+2]^*} \alpha_{i,j} g_{j+8}(t)$ (applying a $(4, 2)$-GC). Using properties of GC-scheme, if master has access to any two among $\{\ell_{0,0}(t), \ldots, \ell_{3,0}(t)\}$, $g_8(t) + g_9(t) + g_{10}(t) + g_{11}(t)$ can be obtained. Similarly, in round-$(t+3)$, if $g_{2i}(t)$ and $g_{2i+1}(t)$ are already available to master, $\mathcal{T}_i(t + 3, 3)$ involves computing $\ell_{i,1}(t) = \sum_{j \in [i:i+2]^*} \alpha_{i,j} g_{j+12}(t)$. Master can recover $g_{12}(t) + g_{13}(t) + g_{14}(t) + g_{15}(t)$ from any two results among $\{\ell_{0,1}(t), \ldots, \ell_{3,1}(t)\}$. This completes the description of the M-SGC scheme. Note that number of data points involved is the same in both fixed and adaptive mini-tasks and hence, computational load remains the same in both situations.

*Analysis of straggler patterns*: We will now show that the scheme tolerates any straggler pattern conforming to the $(B = 2, W = 3, \lambda = 2)$-bursty straggler model. In Fig. 6 (included in the Appendix), we illustrate mini-task assignments with respect to a straggler pattern conforming to this model. As jobs are indexed in the range $[1 : J]$, mini-tasks corresponding to job-$t$, $t \notin [1 : J]$ are indicated using $\mathbf{0}$ in the figure. These are trivial mini-tasks which do not incur any computation. Consider the computation of $g(2)$ (i.e., job-2) based on Fig. 6. Since, worker-0 is a straggler in round-2 and worker-1 is a straggler in rounds 2 and 3, computations of $\{g_0(2), g_2(2), g_3(2)\}$ failed initially, got reattempted and succeeded. From $\ell_{2,0}(2)$ and $\ell_{3,0}(2)$ (both finished in round-4), the master can recover $g_8(2) + g_9(2) + g_{10}(2) + g_{11}(2)$, owing to the use of $(4, 2)$-GC. Similarly, $g_{12}(2) + g_{13}(2) + g_{14}(2) + g_{15}(2)$ can be recovered using $\ell_{0,1}(2)$ and $\ell_{3,1}(2)$ in round-5. Hence, master computes $g(2) \triangleq g_0(2) + \cdots + g_{15}(2)$ after round-5 (delay $T = W - 2 + B = 3$).

### 3.3.2 GENERAL SCHEME

*Data placement*: Assume that dataset $\mathbf{D}$ contains $d$ data points. $\mathbf{D}$ is partitioned into $(W - 1 + B)n$ unequally sized data chunks $\{\mathbf{D}_0, \mathbf{D}_1, \ldots, \mathbf{D}_{(W-1+B)n-1}\}$. Data chunks $\mathcal{D}_1 \triangleq \{\mathbf{D}_0, \mathbf{D}_1, \ldots, \mathbf{D}_{(W-1)n-1}\}$ are all equally sized and contain $\frac{\lambda+1}{n(B+(W-1)(\lambda+1))} d$ data points each. Similarly, $\mathcal{D}_2 \triangleq \{\mathbf{D}_{(W-1)n}, \mathbf{D}_{(W-1)n+1}, \ldots, \mathbf{D}_{(W-1+B)n-1}\}$ are also equally sized and contain $\frac{1}{n(B+(W-1)(\lambda+1))} d$ data points each. Data chunks in $\mathcal{D}_1$ are divided into $n$ groups consisting of $(W - 1)$ data chunks each. Every worker will store one of these groups of data chunks. Precisely, worker-$i$ stores data chunks $\{\mathbf{D}_l\}_{l \in [i(W-1):(i+1)(W-1)-1]}$. Similarly, $\mathcal{D}_2$ is divided into $B$ groups consisting of $n$ data chunks each. Group-$j$, $j \in [0 : B - 1]$, consists of data chunks $\{\mathbf{D}_{(W-1+j)n}, \ldots, \mathbf{D}_{(W+j)n-1}\}$. The $n$ equally sized data chunks in each group will be treated as $n$ partitions of data set in an $(n, \lambda)$-GC scheme (see Sec. 3.1) and will be stored in workers accordingly. i.e., from each group-$j$, worker-$i$ stores the $(\lambda + 1)$ data chunks $\{\mathbf{D}_l\}_{l \in (W-1+j)n+[i:i+\lambda]^*}$.

*Mini-task assignment*: In round-$t$, each worker-$i$ sequentially performs $(W - 1 + B)$ mini-tasks labelled as $\{\mathcal{T}_i(t; 0), \ldots, \mathcal{T}_i(t; W - 2 + B)\}$. A mini-task involves either (i) computing partial gradient with respect to one of the data chunks in $\mathcal{D}_1$ or else (ii) one partial gradient each with respect to $\lambda + 1$ data chunks (thus, $\lambda + 1$ partial gradients in total) in $\mathcal{D}_2$. As noted in the example, failed mini-tasks belonging to scenario (i) will be reattempted whereas those in scenario (ii) will be compensated via use of $(n, \lambda)$-GC scheme. Recall that delay $T = W - 2 + B$. In Algorithm 2, we

describe how mini-tasks are assigned. We have normalized load:

$$L_{\text{M-SGC}} \;=\; \begin{cases} \frac{(\lambda+1)(W-1+B)}{n(B+(W-1)(\lambda+1))}, & \text{if } \lambda < n, \\ \frac{W-1+B}{n(W-1)}, & \text{if } \lambda = n. \end{cases} \quad (1)$$

**Remark 3.2** (Case of $\lambda = n$). For the special case $\lambda = n$, data set $\mathbf{D}$ will be partitioned into $(W-1)n$ equally sized data chunks $\mathcal{D}_1 \triangleq \{\mathbf{D}_0, \mathbf{D}_1, \ldots, \mathbf{D}_{(W-1)n-1}\}$. i.e., effectively we have $\mathcal{D}_2 \triangleq \Phi$ (null set). For notational consistency, we set partial gradients $g_l(t) \triangleq \mathbf{0}, l \in [(W-1)n : (W-1+B)n-1]$. These are trivial partial gradients which do not incur any computation in Algorithm 2.

**Remark 3.3.** It is straightforward to note that for given $\{n, W, B\}$, normalized load is the largest when $\lambda = n$. As $B < W$, we thus have $L_{\text{M-SGC}} \leq \frac{2}{n}$ irrespective of the choice of $\lambda$. In contrast, normalized load $\frac{s+1}{n}$ of SR-SGC scheme scales with $\lambda$ ($s \triangleq \lceil \frac{B\lambda}{W-1+B} \rceil \geq 1$).

---

**Algorithm 2** Algorithm used by master to assign mini-tasks in round-$t$

---

**for** $i \in [0 : n-1]$ **do**
    **for** $j \in [0 : W-2]$ **do**
        Assign computation of $g_{i(W-1)+j}(t-j)$ as mini-task $\mathcal{T}_i(t;j)$.
                ▷ Mini-task result is $g_{i(W-1)+j}(t-j)$
    **for** $j \in [W-1 : W-2+B]$ **do**
        **if** master received all of $\{g_{i(W-1)+j'}(t-j)\}_{j' \in [0:W-2]}$ prior to round-$t$ **then**
            Assign computation of $(\lambda+1)$ partial gradients $\{g_{jn+l}(t-j)\}_{l \in [i:i+\lambda]^*}$ as $\mathcal{T}_i(t;j)$.
                ▷ Mini-task result is $\ell_{i,j-(W-1)}(t-j) = \sum_{l \in [i:i+\lambda]^*} \alpha_{i,l} g_{jn+l}(t-j)$
                        ▷ Coefficients $\alpha_{i,l}$ as defined in Sec. 3.1
        **else**
            **for** $j' \in [0 : W-2]$ **do**
                **if** master has not received $g_{i(W-1)+j'}(t-j)$ prior to round-$t$ **then**
                    Assign computation of $g_{i(W-1)+j'}(t-j)$ as mini-task $\mathcal{T}_i(t;j)$.
                        ▷ Mini-task result is $g_{i(W-1)+j'}(t-j)$
            **break**
                        ▷ breaks the $j'$ for loop

---

**Proposition 3.2.** *Consider the M-SGC scheme designed for parameters $\{n, B, W, \lambda\}$, where $0 \leq \lambda \leq n, 0 < B < W$, $T = W - 2 + B$ and $L_{\text{M-SGC}}$ as in equation 1. The scheme tolerates any straggler pattern which conforms to either the $(B, W, \lambda)$-bursty straggler model or else, the $(N = B, W' = W + B - 1, \lambda' = \lambda)$-arbitrary straggler model.*

**Remark 3.4** (Near-optimality). Based on information-theoretic bounds (see Appendix F for more details), we observe that when $\lambda = n - 1$ or $n$, M-SGC scheme is optimal as a scheme tolerating any straggler pattern conforming to the $(B, W, \lambda)$-bursty straggler model. Moreover, for fixed $n, B$ and $\lambda$, the gap between the proposed load in equation 1 and optimal load decreases as $O(\frac{1}{W})$. Analogous results are derived with respect to the $(N, W', \lambda')$-arbitrary straggler model as well.

**Remark 3.5.** If $(s+1)$ divides $n$, there exists a simplification to the GC scheme. Both SR-SGC and M-SGC schemes can leverage the existence of such a simplified scheme. We discuss this in App. G.

## 4 EXPERIMENTAL RESULTS

In this section, we evaluate the performance of proposed schemes by training multiple neural network models concurrently. We use the AWS Lambda, a fully-managed and cost-efficient serverless cloud computing service. Workers are invoked from the master node using HTTP requests, and task results are received in the HTTP response payload. Appendices H and I provide detailed discussions on the network architecture, experimental setup and potential applications.

### 4.1 ANALYSIS OF RESPONSE TIME

Our experiment setup consists of a master node and $n = 256$ workers. In Fig. 1, we demonstrate statistics of response time across 100 rounds, where each worker calculates gradients for a batch of 16 MNIST images on a CNN involving three convolutional layers, followed by two fully connected

layers. Fig. 1(a) shows the response time of each worker at every round. White cells represent stragglers. As discussed in Sec. 2, a worker is deemed straggler when its response time exceeds $(1 + \mu)$ times the response time of the fastest worker in the round. For the sake of consistency, we choose $\mu = 1$ for all experiments. Nonetheless, such a choice of $\mu$ is by no means critical to observe stragglers. This can be seen in Fig. 1(c), where the empirical CDF of workers' completion time exhibits a relatively long tail. Fig. 1(b) plots the number of straggler bursts of different length over this response profile. It can be observed that our response profile does not include nodes that continue to remain stragglers for long duration. This motivates the use of coding across the temporal dimension as proposed in the present work.

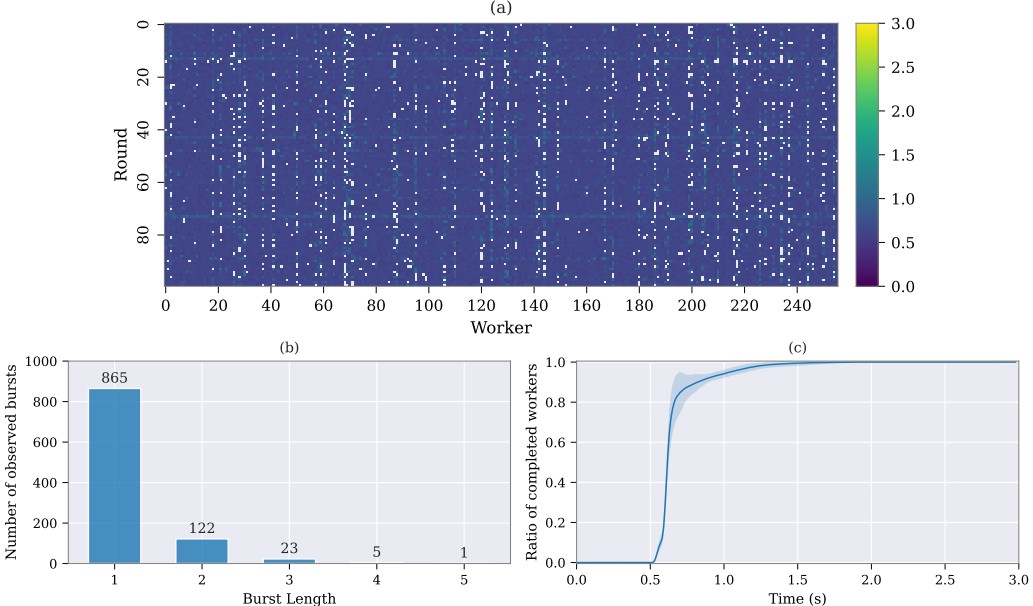

Figure 1: Statistics of response time for 256 workers across 100 rounds. Each worker calculates gradients of the loss for a batch of 16 MNIST images on a convolutional neural network. (a) Each white cell represents a worker (x-axis) who is a straggler at the corresponding round (y-axis). (b) Histogram of stragglers' burst lengths. (c) Empirical CDF of workers' completion time, averaged over 100 rounds (shades represent standard deviation).

## 4.2 COMPARISON OF CODING SCHEMES

Using the setup described in Sec. 4.1, we train $M = 4$ CNN classifiers for MNIST concurrently following the approach stated in Remark 2.1. In every round, master samples a batch of $4096$ data points and distributes them among the workers. Non-straggling workers compute partial gradients and return task results to the master at the end of each round. After completion of one update, master uploads the updated model parameters to a shared network file system, accessible to the workers. We use cross entropy as the loss function and ADAM as the optimizer. Moreover, the same dataset and architecture are used for all the models.

In each experiment, we run a total of $J = 480$ jobs (120 jobs per classifier) using the three schemes, namely GC, SR-SGC and M-SGC. As a baseline, we also train the classifiers without any coding wherein the master node should wait for all the workers to return their task results. Finally, each experiment is repeated 10 times to report the first and second-order statistics of total run times. Before training the models, we perform some shorter experiments to choose the best-performing parameters for each of the three coding schemes. Specifically, for GC, we perform a grid search over $s$ and select the value corresponding to the shortest run time. We refer readers to Appendix J for a detailed discussion on the procedure of selecting the parameters for SR-SGC and M-SGC schemes, as well as analysis of sensitivity to parameters.

Table 1 presents the total run time achieved by each coding scheme, along with the selected parameters and resulting normalized loads. Selection of small values for parameters $B$ and $W$ in our sequential

coding schemes matches the empirical evidence in Fig. 1(b) that isolated short-length-bursts are prevalent. It is interesting to note that the effective value of parameter $s$ in SR-SGC ($s = 12$) turns out to be close to that of GC ($s = 15$). Fig. 2(a) plots total number of completed jobs (for all $M = 4$ models) across time, and Fig. 2(b) shows the course of training loss (of the first model out of the 4 models) as a function of time, for all coding schemes.

Table 1: Total run time achieved by different coding schemes

| Scheme | Parameters | Normalized Load | Run Time (s) |
|--------|-----------|-----------------|--------------|
| M-SGC | $B = 1, W = 2, \lambda = 27$ | 0.008 | $891.37 \pm 43.10$ |
| SR-SGC | $B = 2, W = 3, \lambda = 23 \;\; (s = 12)$ | 0.051 | $994.22 \pm 43.66$ |
| GC | $s = 15$ | 0.062 | $1064.96 \pm 46.72$ |
| No Coding | $-$ | 0.004 | $1307.79 \pm 61.88$ |

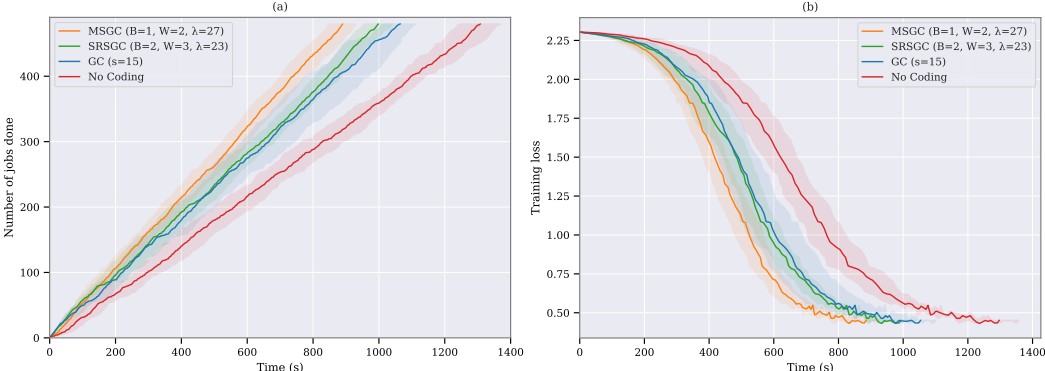

Figure 2: (a) Number of completed rounds vs. clock time, averaged over 10 independent experiments. (b) Training loss vs. clock time for the first model (out of four concurrently trained models), averaged over 10 independent runs. Shades here represent standard deviation.

The first clear observation from Table 1 is that our proposed M-SGC achieves 16% lower run time while maintaining smaller normalized load compared to the classical GC scheme. Furthermore, compared to GC, SR-SGC shows slight improvements in total runtime and normalized load simultaneously, demonstrating the potential of incorporating selective repetition into GC. Next, as shown in Fig. 2 and Table 1, the existence of stragglers is validated by the fact that any of the coding schemes significantly outperforms the case of not using any coding. This is indeed in line with the empirical observation of Figure 1(c), where the tail of the cumulative distribution of workers' completion time signals the existence of stragglers. Appendix K discusses how the overheads of decoding time and parameter selection time can be completely removed in the training process. In Appendix L, we present analogous results for concurrently training four ResNet-18 models on CIFAR-100 dataset.

## 5 CONCLUSION

We develop a new class of gradient coding schemes that exploit coding across the temporal dimension, in addition to the spatial dimension (i.e., coding across workers) considered in prior works. Our first scheme, SR-SGC uses the GC scheme in Tandon et al. (2017) as a base scheme and enhances it by performing selective repetition of tasks based on the past sequence of straggler patterns. Our second scheme, M-SGC multiplexes gradient coding and selective repetition of tasks over the dataset in a novel way that dramatically reduces the computational load per worker, when compared to both GC and SR-SGC. In addition, we demonstrate that the computational load of M-SGC can approach an information theoretic lower bound in certain cases. We validate our schemes through experiments over a large scale AWS Lambda cluster involving 256 worker nodes. We first analyze the response time of workers and provide empirical evidence of straggler patterns that are consistent with our modeling assumptions. We then demonstrate that the M-SGC scheme provides significant gains over all the other schemes in real world experiments.

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

## A  SUMMARY OF NOTATIONS

Table 2: Table of notations

| Symbol | Meaning |
|---|---|
| $n$ | Number of workers |
| $J$ | Number of jobs |
| $T$ | Delay parameter |
| $\mathbb{Z}$ | The set of all integers |
| $[a:b]$ | $\{i \in \mathbb{Z} \mid a \le i \le b\}$ |
| $[a:b]^*$ | $\{i \mod n \mid i \in [a:b]\}$ |
| $c + [a:b]^*$ | $\{c + i \mid i \in [a:b]^*\}$ |
| $\mathbf{D}$ | Dataset |
| $\mathbf{D}_i$ | $i$-th data chunk |
| $g(t)$ | Gradient corresponding to job-$t$ |
| $g_i(t)$ | $i$-th partial gradient corresponding to job-$t$ |
| $\eta$ | Number of data chunks |
| $\gamma_i(t)$ | Number of partial gradients computed by worker-$i$ in round-$t$ |
| $s$ | Upper limit on the number of stragglers per round (GC, SR-SGC schemes) |
| $\mu$ | Tolerance parameter |
| $L_i(t)$ | Normalized computational load at worker-$i$ in round-$t$ |
| $W$ | Window size for the bursty straggler model |
| $W'$ | Window size for the arbitrary straggler model |
| $\lambda$ | Upper limit on the number of stragglers in any window (bursty straggler model) |
| $\lambda'$ | Upper limit on the number of stragglers in any window (arbitrary straggler model) |
| $B$ | Upper limit on the length of burst |
| $N$ | Upper limit on the number of straggling rounds of a worker in any window |
| $\ell_i(t)$ | Task result corresponding to job-$t$ returned by worker-$i$ (GC, SR-SGC schemes) |
| $\mathcal{T}_i(t; j)$ | $j$-th mini-task of worker-$i$ in round-$t$ (M-SGC scheme) |
| $\ell_{i,j}(t)$ | Task result corresponding to job-$t$ returned by worker-$i$ (M-SGC scheme) |

## B  A BRIEF OVERVIEW ON THE USE OF CODING FOR STRAGGLER MITIGATION IN DISTRIBUTED COMPUTING SYSTEMS

Stragglers, or slow processing workers, are common in distributed computing systems and potentially delay the timely completion of computational jobs. One may use the following analogy to understand the importance of erasure codes in the context of distributed computing. Consider an $[n, k]$ erasure code which encodes $k$ bits of data to produce $n > k$ coded bits for transmission over a channel which could erase a few bits. Even if part of the $n$ coded bits are erased by the channel in this case, the redundant $(n - k)$ bits aid in data recovery. The final job result to be obtained by the master in a distributed computing system may be compared to the $k$-bit data which is being encoded. Computations undertaken by a worker may be thought of as a coded bit of an erasure code. Naturally, delayed responses due to stragglers correspond to bit erasures. The core idea behind existing works in the literature is that erasure codes offer a way to efficiently add redundancy to computations or data so that jobs can still be completed on schedule, even when some computations are unavailable. The use of erasure coding to combat stragglers in distributed computing systems is initially investigated in the work Lee et al. (2018) (in the context of distributed matrix multiplication). The survey Li & Avestimehr (2020) provides a comprehensive summary of recent developments in the area.

Consider the following toy example to demonstrate the key idea employed in Lee et al. (2018). Assume that the master node wants to distribute the job of computing $A^\top x$ across $n = 3$ workers, where any one of the workers could be a straggler. The master will partition the columns of $A$ as $[A_0 \mid A_1]$. It will now pass $\{A_0, x\}$, $\{A_1, x\}$ and $\{A_2 \triangleq (A_0 + A_1), x\}$ to workers 0, 1 and 2, respectively. Worker-$i$ will attempt to compute $A_i^\top x$ and return the result. Clearly, the master can compute $A^\top x$ as soon as it receives results from any two workers and hence the system is resilient against one straggler.

The gradient coding framework proposed in Tandon et al. (2017) examines the possibility of using erasure codes to perform distributed stochastic gradient descent in a straggler-resilient manner. We borrow the following example from the original paper to highlight the key idea. Let the dataset $\mathbf{D}$ be partitioned into $\mathbf{D}_0$, $\mathbf{D}_1$ and $\mathbf{D}_2$. Workers 0, 1 and 2 store $\{\mathbf{D}_0, \mathbf{D}_1\}$, $\{\mathbf{D}_1, \mathbf{D}_2\}$ and $\{\mathbf{D}_2, \mathbf{D}_0\}$, respectively. Worker-0 is supposed to compute partial gradients $\{g_0, g_1\}$ on data chunks $\{\mathbf{D}_0, \mathbf{D}_1\}$ and then return $\ell_0 \triangleq \frac{g_0}{2} + g_1$ to the master. Similarly, workers 1 and 2 attempt to return $\ell_1 \triangleq g_1 - g_2$ and $\ell_2 \triangleq \frac{g_0}{2} + g_2$, respectively. It can be inferred that $g \triangleq g_0 + g_1 + g_2$ can be computed from the results returned by any two workers. For instance, suppose workers 0 and 1 have returned their results and worker-2 is a straggler. In this case, we have $g = 2(\frac{g_0}{2} + g_1) - (g_1 - g_2) = 2\ell_0 - \ell_1$.

## C    ON THE USE OF GILBERT-ELLIOT MODEL FOR STRAGGLERS

The Gilbert-Elliot (GE) model is a 2-state probabilistic model with state transition probabilities as outlined in Fig. 3. In the context of modeling stragglers, a worker is a straggler if and only if it is in state $\mathcal{S}$ and a non-straggler otherwise. If a worker is a straggler in a given round, it will continue to be a straggler in the next round with probability $(1 - p_{\mathcal{S}})$. Similarly, a non-straggler will continue to remain so in the next round with probability $(1 - p_{\mathcal{N}})$. In essence, the GE model suggests that straggling behavior occurs periodically and is often followed by non-straggling behavior.

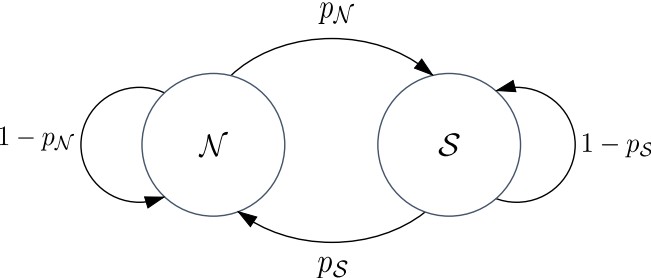

Figure 3: The 2-state Gilbert-Elliot model.

In distributed systems, stragglers occur due to reasons such as resource sharing, communication bottlenecks, routine maintenance activities etc. The periodic spikes in latency that are caused by several of these factors (see Dean & Barroso (2013); Bolot (1993)) are naturally captured by the GE model. The authors of Yang et al. (2019) make an empirical observation that the GE model can accurately track the state transitions of workers on an Amazon EC2 cluster, particularly in the context of applying erasure codes for straggler mitigation. In the current study, we approximate the GE model using deterministic, sliding-window-based models since they are more tractable in terms of code design. As our experiments indicate, such a design strategy finally led to techniques that outperform the baseline GC on an Amazon Lambda cluster.

We will now present an intuitive justification on why sliding-window-based models are a natural candidate to model straggler behavior in our work. Note that the sliding-window-based models introduce constraints on the local structure of straggler patterns. For example, in the bursty model, a burst of straggling rounds is followed by a period when the worker is a non-straggler. This provides a simplified approximation of the GE model, capturing the dominant set of straggler patterns associated with the GE channel (any other straggler pattern will be handled through wait-outs). Similarly, in the arbitrary straggler model, we put a constraint on the total number of straggling rounds in each window without requiring that the straggling rounds be consecutive. Finally, in our proposed setting of sequential gradient coding, it turns out that modeling local constraints on straggler patterns using the sliding-window-straggler model is a natural fit – for a job that starts in round-$i$ and is completed in round-$(i + \Delta)$, the straggler patterns around the interval $[i : i + \Delta]$ are relevant.

## D    PROOF OF PROPOSITION 3.1

Consider a straggler pattern which conforms to either (i) the $(B, W, \lambda)$-bursty straggler model or else, (ii) the $s$-stragglers-per-round model, in any window of rounds $W_j \triangleq [j : j + W - 1]$, $j \in [1 : J + T - W + 1]$. Let $t' \in [1 : J + T]$ denote the first round in which there are more than

$s$ stragglers. All the tasks attempted in round-$t'$ correspond to job-$t'$. By assumption, the straggler pattern, when restricted to the $W$ rounds $[t' : t' + xB]$ conforms to the $(B, W, \lambda)$-bursty straggler model. Let there be $\lambda_0 > s$ stragglers in round-$t'$. In round-$(t' + B)$, $\lambda_0 - s$ tasks corresponding to job-$t'$ will be attempted by $\lambda_0 - s$ workers who were stragglers in round-$t'$ (see Fig. 4). Because of the straggler model assumption, none of these workers will be stragglers again in round-$(t' + B)$ and hence, job-$t'$ will be finished in round-$(t' + B)$ with delay $B$. Clearly, round-$(t' + B)$ is a "deviation" from the $(n, s)$-GC scheme as not all tasks in this round correspond to job-$(t' + B)$. Suppose now there are $\lambda_1$ stragglers in round-$(t' + B)$. Thus, number of task results corresponding to job-$(t' + B)$ returned by workers in round-$(t' + B)$ is given by $n - \lambda_1 - \lambda_0 + s$. If this quantity is greater than or equal to $n - s$, job-$(t' + B)$ can be finished in round-$(t' + B)$ itself and there will be a "reset" in round-$(t' + 2B)$ as all tasks there correspond to job-$(t' + 2B)$. On the other hand, if $(n - \lambda_1 - \lambda_0 + s) < (n - s)$, $(\lambda_0 + \lambda_1 - 2s)$ tasks corresponding to job-$(t' + B)$ will be attempted in round-$(t' + 2B)$ by workers who did not return task results corresponding to job-$(t' + B)$ in round-$(t' + B)$. These workers can either be stragglers in round-$(t' + B)$ or else, be processing tasks corresponding to job-$t'$ in round-$(t' + B)$. In either case, these workers cannot be stragglers in round-$(t' + 2B)$ owing to the straggler model (see Fig. 4). Hence, job-$(t' + B)$ will finish in round-$(t' + 2B)$ will delay $B$. Now, if not enough task results corresponding to job-$(t' + 2B)$ are returned in round-$(t' + 2B)$, the minimum-required number of additional tasks corresponding to this job will be attempted in round-$(t' + 3B)$ and so on. We will now show that there exist an $\ell \in [1 : x]$ such that jobs $t', t' + B, \ldots, t' + (\ell - 1)B$ are finished with delay precisely $B$ and job-$(t' + \ell B)$ is finished with delay 0. i.e., there is a reset happening in round-$(t' + (\ell + 1)B)$. In order to show this, assume that jobs $t', t' + B, \ldots, t' + (x - 1)B$ have some of their tasks attempted with delay $B$ (i.e., no reset in rounds $t' + 2B, \ldots, t' + xB$). Because of the straggler model, all these delayed tasks are guaranteed to succeed and hence all these jobs finish with delay $B$. We should now prove that there is a reset in round-$(t' + (x + 1)B)$, i.e., $\ell = x$. For $j \in [0 : x]$, let $\lambda_j$ indicate the number of stragglers in round-$(t' + jB)$. Number of tasks corresponding to job-$(t' + (x - 1)B)$ attempted in round-$(t' + xB)$ is given by $\lambda_0 + \lambda_1 + \cdots + \lambda_{x-1} - xs$. Thus, number of task results corresponding to job-$(t' + xB)$ received by master in round-$(t' + xB)$ is given by:

$$
\begin{aligned}
n - \lambda_x - (\lambda_0 + \lambda_1 + \cdots + \lambda_{x-1} - xs) &= n - \lambda_0 - \lambda_1 - \cdots - \lambda_x + xs \\
&\geq n - \lambda + xs \\
&\geq n - \lceil \frac{\lambda}{x+1} \rceil (x + 1) + xs \\
&= n - s(x + 1) + xs \\
&= n - s,
\end{aligned}
$$

where we have used the fact that $\lambda_0 + \cdots + \lambda_x \leq \lambda$ owing to the straggler model. Hence, in summary, we have showed that if all of jobs $t', t' + B, \ldots, t' + (x - 1)B$ finish with delay $B$, then $t' + xB$ finishes with delay 0 and there will be a reset in round-$(t' + (x + 1)B)$. Thus, there exist $\ell \in [1 : x]$ such that all jobs $t', t' + B, \ldots, t' + (\ell - 1)B$ finish with delay $B$ and job-$(t' + \ell B)$ finishes with delay 0. Because of the reset happening in round-$(t' + (\ell + 1)B)$, the "effect" of Algorithm 1 is now confined only to rounds $t', t' + B, \ldots, t' + \ell B$. We can now safely regard rounds $t', t' + B, \ldots, t' + \ell B$ as straggler-free as these rounds contain only tasks corresponding to jobs $t', t' + B, \ldots, t' + \ell B$ and we have shown that all these jobs succeed with delay at most $B$. We can now essentially repeat all these steps starting with finding the next "first" round-$t'$ having more than $s$ stragglers. After repeating these arguments sufficient number of times, eventually, we will be left with jobs $R \subseteq [1 : J]$, where all rounds in $R$ has at most $s$ stragglers. Workers in each round-$r$, $r \in R$, attempt only tasks corresponding to job-$r$. Thus, all these jobs can be finished with delay 0. This completes the proof.

## E  PROOF OF PROPOSITION 3.2

Consider the computation of $g(t)$, $t \in [1 : J]$ in the presence of a straggler pattern conforming to one of the following; (i) $(B, W, \lambda)$-bursty straggler model or else, (ii) $(N = B, W' = W + B - 1, \lambda' = \lambda)$-arbitrary straggler model. By design of Algorithm 2, for each worker-$i$, mini-tasks $\{\mathcal{T}_i(t; 0), \mathcal{T}_i(t + 1; 1), \ldots, \mathcal{T}_i(t + W - 2 + B; W - 2 + B)\}$ correspond to job-$t$. Master has to

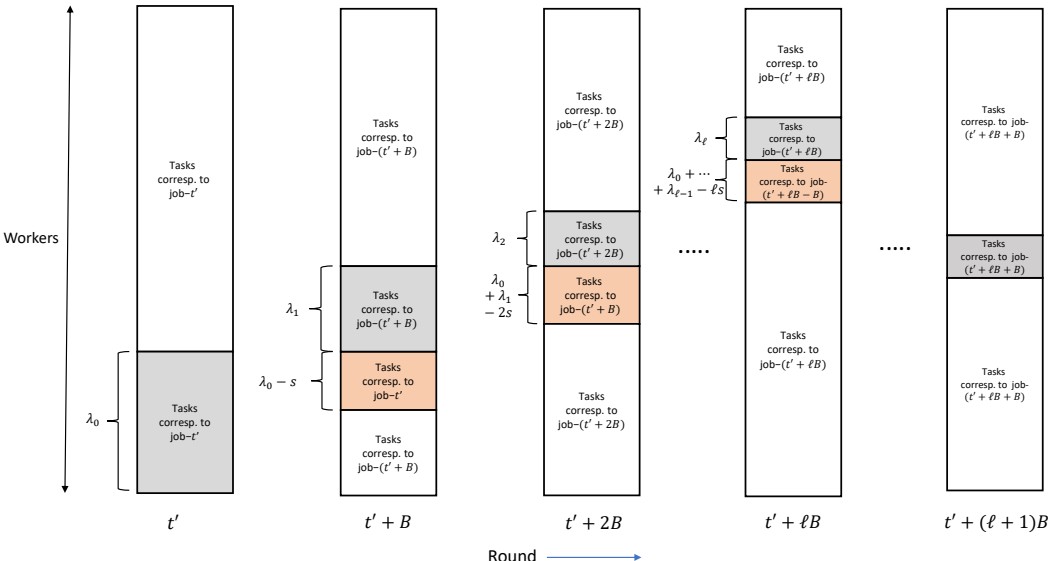

Figure 4: An illustration of task assignment in SR-SGC. In the initial rounds, the task assignment is precisely as in an $(n, s)$-GC scheme and all tasks in round-$t$ correspond to job-$t$. In any such round-$t$, if there are at most $s$ stragglers, job-$t$ can be finished in round-$t$ itself. Let $t'$ denote a round where all tasks correspond to job-$t'$ and there are $\lambda_0 > s$ stragglers. Stragglers are indicated in grey color. In round-$(t' + B)$, there is a deviation from the $(n, s)$-GC scheme as $\lambda_0 - s$ tasks corresponding to job-$t'$ will be attempted in this round (these tasks which are attempted with delay $B$ are indicated in orange). These tasks are attempted by workers who did not return task results corresponding to job-$t'$ in round-$t'$. In round-$(t' + B)$, $n - \lambda_1 - \lambda_0 + s$ workers return task results corresponding to job-$(t' + B)$. If this quantity is lesser than $n - s$, $\lambda_0 + \lambda_1 - 2s$ tasks corresponding to job-$(t' + B)$ will be attempted in round-$(t' + 2B)$ by workers who did not return task results corresponding to job-$(t' + B)$ in round-$(t' + B)$. The process is continued in a similar manner. If the straggler pattern in the window of rounds $[t' : t' + xB]$ conforms to the $(B, W, \lambda)$-bursty straggler model, there exists an $\ell \in [1 : x]$ such that in round-$(t' + (\ell + 1)B)$ a "reset" happens back to the $(n, s)$-GC scheme, i.e., in round-$(t' + (\ell + 1)B)$ all tasks correspond to job-$(t' + (\ell + 1)B)$.

Round $\longrightarrow$

| $\mathcal{T}_i(5; 0)$ | $\mathcal{T}_i(6; 0)$ | $\mathcal{T}_i(7; 0)$ | $\mathcal{T}_i(8; 0)$ | $\mathcal{T}_i(9; 0)$ | $\mathcal{T}_i(10; 0)$ |
|---|---|---|---|---|---|
| $\mathcal{T}_i(5; 1)$ | $\mathcal{T}_i(6; 1)$ | $\mathcal{T}_i(7; 1)$ | $\mathcal{T}_i(8; 1)$ | $\mathcal{T}_i(9; 1)$ | $\mathcal{T}_i(10; 1)$ |
| $\mathcal{T}_i(5; 2)$ | $\mathcal{T}_i(6; 2)$ | $\mathcal{T}_i(7; 2)$ | $\mathcal{T}_i(8; 2)$ | $\mathcal{T}_i(9; 2)$ | $\mathcal{T}_i(10; 2)$ |
| $\mathcal{T}_i(5; 3)$ | $\mathcal{T}_i(6; 3)$ | $\mathcal{T}_i(7; 3)$ | $\mathcal{T}_i(8; 3)$ | $\mathcal{T}_i(9; 3)$ | $\mathcal{T}_i(10; 3)$ |

Mini-tasks corresp. to job-5

Figure 5: Figure corresponding to the M-SGC example provided in Sec. 3.3.1. For an M-SGC scheme, all mini-tasks across a "diagonal" correspond to the same job.

| | | | | | |
|---|---|---|---|---|---|
| $g_0(1)$ | $g_0(2)$ | $g_0(3)$ | $g_0(4)$ | $g_0(5)$ | $g_0(6)$ |
| **0** | $g_1(1)$ | $g_1(2)$ | $g_1(3)$ | $g_1(4)$ | $g_1(5)$ |
| **0** | **0** | $\boldsymbol{g_0(1)}$ | $\boldsymbol{g_0(2)}$ | $\ell_{0,0}(3)$ | $\ell_{0,0}(4)$ |
| **0** | **0** | **0** | $\boldsymbol{g_1(1)}$ | $\ell_{0,1}(2)$ | $\ell_{0,1}(3)$ |

Worker-0

| | | | | | |
|---|---|---|---|---|---|
| $g_2(1)$ | $g_2(2)$ | $g_2(3)$ | $g_2(4)$ | $g_2(5)$ | $g_2(6)$ |
| **0** | $g_3(1)$ | $g_3(2)$ | $g_3(3)$ | $g_3(4)$ | $g_3(5)$ |
| **0** | **0** | $\boldsymbol{g_3(1)}$ | $\boldsymbol{g_2(2)}$ | $\boldsymbol{g_2(3)}$ | $\ell_{1,0}(4)$ |
| **0** | **0** | **0** | $\boldsymbol{g_3(1)}$ | $\boldsymbol{g_3(2)}$ | $\ell_{1,1}(3)$ |

Worker-1

| | | | | | |
|---|---|---|---|---|---|
| $g_4(1)$ | $g_4(2)$ | $g_4(3)$ | $g_4(4)$ | $g_4(5)$ | $g_4(6)$ |
| **0** | $g_5(1)$ | $g_5(2)$ | $g_5(3)$ | $g_5(4)$ | $g_5(5)$ |
| **0** | **0** | $\ell_{2,0}(1)$ | $\ell_{2,0}(2)$ | $\ell_{2,0}(3)$ | $\boldsymbol{g_5(4)}$ |
| **0** | **0** | **0** | $\ell_{2,1}(1)$ | $\ell_{2,1}(2)$ | $\ell_{2,1}(3)$ |

Worker-2

| | | | | | |
|---|---|---|---|---|---|
| $g_6(1)$ | $g_6(2)$ | $g_6(3)$ | $g_6(4)$ | $g_6(5)$ | $g_6(6)$ |
| **0** | $g_7(1)$ | $g_7(2)$ | $g_7(3)$ | $g_7(4)$ | $g_7(5)$ |
| **0** | **0** | $\ell_{3,0}(1)$ | $\ell_{3,0}(2)$ | $\ell_{3,0}(3)$ | $\ell_{3,0}(4)$ |
| **0** | **0** | **0** | $\ell_{3,1}(1)$ | $\ell_{3,1}(2)$ | $\ell_{3,1}(3)$ |

Worker-3

| 1 | 2 | 3 | 4 | 5 | 6 |

Round $\longrightarrow$

Figure 6: Figure corresponding to the M-SGC example provided in Sec. 3.3.1. Rectangles depict mini-tasks (shaded ones have failed due to stragglers). Reattempted mini-tasks are indicated in red. Mini-task results in blue are linear combinations of 3 partial gradients.

compute:

$$g(t) \triangleq \underbrace{\sum_{j \in [0:(W-1)n-1]} g_j(t)}_{g'(t)} + \underbrace{\sum_{l \in [(W-1)n:(W-1+B)n-1]} g_l(t)}_{g''(t)}$$

by the end of round-$(t + T)$, where $T = W - 2 + B$. If $\lambda = n$, we only have the $g'(t)$ part and set $g''(t) \triangleq 0$. We will now show that master will be able to compute each of $\{g'(t), g''(t)\}$ individually by the end of round-$(t + W - 2 + B)$ in presence of straggler patterns conforming to one of these straggler models.

*Computing $g'(t)$:* From Algorithm 2, it can be noted that mini-task $\mathcal{T}_i(t + j; j), j \in [0 : W - 2]$ involves computing $g_{i(W-1)+j}(t)$. If worker-$i$ is not a straggler in all the rounds $[t : t + W - 2]$, clearly, master can compute $\sum_{j \in [0:W-2]} g_{i(W-1)+j}(t)$ in the end of round-$(t + W - 2)$.

Now, consider the remaining situation that worker-$i$ is a straggler in at least one of the rounds within $[t : t + W - 2]$. We initially discuss the case that the straggler pattern conforms to $(B, W, \lambda)$-bursty straggler model. Worker-$i$ experiences at most $B$ straggling rounds (see Fig. 7) among rounds $[t : t + W - 2 + B]$. Suppose worker-$i$ is a straggler in $x'$ rounds, $x' \in [1 : B]$, within rounds $[t : t + W - 2]$. Thus, $x'$ partial gradients among $\{g_{i(W-1)+j}(t)\}_{j \in [0:W-2]}$ are not returned by worker-$i$ in rounds $[t : t + W - 2]$. However, Algorithm 2 reattempts those failed partial gradient computations in rounds $[t + W - 1 : t + W - 2 + B]$. Even if there are $B - x'$ straggling rounds among $[t + W - 1 : t + W - 2 + B]$, in the remaining $x'$ rounds, the failed partial gradients will be successfully computed. Hence, using mini-task results returned by worker-$i$, master can compute $\sum_{j \in [0:W-2]} g_{i(W-1)+j}(t)$ by the end of round-$(t + W - 2 + B)$. By accumulating results from all the $n$ workers, master will be able to compute $g'(t)$ by the end of round-$(t + W - 2 + B)$.

We now consider the case where straggler pattern conforms to the $(N = B, W' = W + B - 1, \lambda' = \lambda)$-arbitrary straggler model. As per the model, a worker can be a straggler in at most $N = B$ rounds in any sliding window consisting of $W' = W - 1 + B$ consecutive rounds. Let worker-$i$ be a straggler in $x''$ rounds ($x'' \in [1 : B]$) within rounds $[t : t + W - 2]$ and at most $B - x''$ rounds within $[t + W - 1 : t + W - 2 + B]$. Clearly, $x''$ mini-tasks among $\{g_{i(W-1)+j}(t)\}_{j \in [0:W-2]}$ fail in their first attempt. However, as worker-$i$ is a non-straggler in at least $x''$ rounds within

$[t + W - 1 : t + W - 2 + B]$, these mini-tasks will eventually be repeated and finished. Thus, master computes $\sum_{j \in [0:W-2]} g_{i(W-1)+j}(t)$ by the end of round-$(t + W - 2 + B)$. Collecting results from all the $n$ workers, master will be able to compute $g'(t)$ by the end of round-$(t + W - 2 + B)$.

*Computing $g''(t)$*: Again, we begin with discussing the case where a straggler pattern conforms to the $(B, W, \lambda)$-bursty straggler model. For any straggler pattern conforming to this straggler model, there exist $(n-\lambda)$ workers who do not have any straggling rounds among any window of rounds of the form $W_j \triangleq [j : j + W - 1]$. In particular, consider the rounds in $W_t$. Assume that worker-$i$ does not have any straggling rounds in the window $W_t$. Thus, mini-tasks $\{\mathcal{T}_i(t + l; l)\}_{l=0}^{W-2}$ will be successful and partial gradients $\{g_{i(W-1)+l}(t)\}_{l=0}^{W-2}$ will be computed in the first attempt. Hence, from Algorithm 2, it can be inferred that the mini-task $\mathcal{T}_i(t + W - 1; W - 1)$ involves computing $\ell_{i,0}(t)$. As worker-$i$ is not a straggler in round-$(t + W - 1)$, master receives the linear combination $\ell_{i,0}(t)$. As there are $(n - \lambda)$ such workers returning $\ell_{i,0}(t)$'s, owing to the use of $(n, \lambda)$-GC, master can compute $\sum_{l \in [(W-1)n:Wn-1]} g_l(t)$ by the end of round-$(t + W - 1)$. Similarly, due to the $(B, W, \lambda)$-bursty straggler model assumption, there are $(n - \lambda)$ workers who do not have any straggling rounds in the window $W_{t+1} = [t + 1 : t + W]$. Let worker-$i'$ be one such worker. All the mini-tasks $\{\mathcal{T}_{i'}(t + 1; 1), \cdots, \mathcal{T}_{i'}(t + W - 2; W - 2)\}$ will be successful and $\{g_{i'(W-1)+l}(t)\}_{l=1}^{W-2}$ will be computed in their first attempts. In round-$t$, worker-$i'$ can possibly be a straggler. However, as worker-$i'$ is not a straggler in round-$(t + W - 1)$, the failed computation of $g_{i'(W-1)}(t)$ will be reattempted and finished in round-$(t + W - 1)$. Thus, by the end of round-$(t + W - 1)$, all partial gradients $\{g_{i'(W-1)+l}(t)\}_{l=0}^{W-2}$ are guaranteed to be computed. Hence, mini-task $\mathcal{T}_{i'}(t + W; W)$ involves computing $\ell_{i',1}(t)$. As round-$(t + W)$ is a non-straggling round, master will receive $\ell_{i',1}(t)$ in the end of round-$(t + W)$. Using $(n - \lambda)$ such $\ell_{i',1}(t)$'s, master can compute $\sum_{l \in [Wn:(W+1)n-1]} g_l(t)$. In a similar manner, it can be argued that, for $m \in [0 : B - 1]$, master will be able to compute $\sum_{l \in [(W-1+m)n:(W+m)n-1]} g_l(t)$ in the end of round-$(t + W - 1 + m)$. Hence, by the end of round-$(t + W - 2 + B)$, master is able to compute $g''(t)$.

In the case if straggler pattern conforms to the $(N = B, W' = W + B - 1, \lambda' = \lambda)$-arbitrary straggler model, there are $n - \lambda$ workers who do not have any straggling rounds in $[t : t+W-2+B]$. For any such worker-$i$, computations of $\{g_{i'(W-1)+l}(t)\}_{l=0}^{W-2}$ are all finished by the end of round-$(t + W - 2)$. Hence, in round-$(t + j)$, $j \in [W - 1 : W - 2 + B]$, worker-$i$ will compute and return $\ell_{i,j-W+1}(t)$ to master. Using $(n, \lambda)$-GC, results from $n - \lambda$ such workers can be used by master to obtain $\sum_{l \in [jn:(j+1)n-1]} g_l(t)$. Thus, by the end of round-$(t + W - 2 + B)$, master is able to compute $g''(t)$.

## F   NEAR-OPTIMALITY OF M-SGC SCHEME

As seen earlier, SR-SGC scheme offers a clear advantage over GC scheme, as it tolerates a super set of straggler patterns; bursty and $s$-stragglers-per-round straggler patterns, for the same normalized

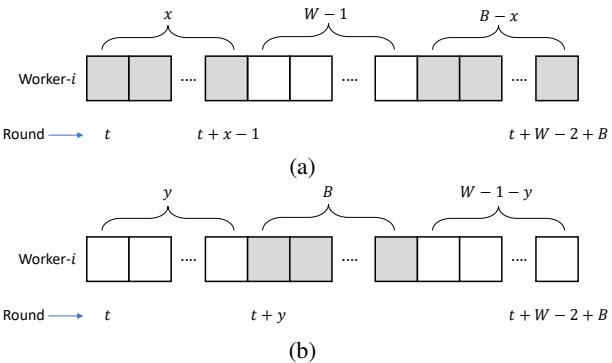

Figure 7: In the figure, shaded boxes depict straggling rounds. Consider a straggler pattern conforming to the $(B, W, \lambda)$-bursty straggler model. The straggling rounds, if any, seen by worker-$i$ among rounds $[t : t + W - 2 + B]$ will be a subset of the $B$ straggling rounds indicated in either situation (a) or (b). Here $x \in [1 : B]$, $y \in [0 : W - 1]$.

load of $\frac{s+1}{n}$. In the case of M-SGC scheme, the normalized load is substantially smaller as we note in Remark 3.3. We will now show that the M-SGC scheme is near-optimal in terms of normalized load. In the theorem below, we derive a fundamental lower bound on normalized load of any sequential gradient coding scheme which tolerates straggler patterns conforming to the $(B, W, \lambda)$-bursty straggler model.

**Theorem F.1.** *Consider any sequential gradient coding scheme, which tolerates straggler patterns permitted by the $(B, W, \lambda)$-bursty straggler model. Let the normalized load per worker $L$ be a constant, $T < \infty$ and the number of jobs $J \to \infty$. We have the following:*

$$L \geq L_B^* \quad \triangleq \quad \begin{cases} \frac{W-1+B}{n(W-1)+B(n-\lambda)}, & \text{if } B < W, \\ \frac{1}{n-\lambda}, & \text{if } B = W. \end{cases} \tag{2}$$

*Proof.* We divide the proof into two cases; $B < W$ and $B = W$.

- $B < W$: Consider the periodic straggler pattern shown in Fig. 8, which conforms to the $(B, W, \lambda)$-bursty straggler model. Consider now the first $\eta(W - 1 + B)$ rounds where $\eta > 0$ is a large enough integer. There are $\eta B \lambda$ straggling rounds faced by workers $[0 : \lambda - 1]$ among these $\eta(W - 1 + B)$. The workers can compute at most gradients $\{g(t)\}_{t \in [1:\eta(W-1+B)]}$ in these rounds. However, the computations that workers can perform is also limited by the normalized load $L$. Specifically, if a worker has normalized load $L$, it is able to compute $L$ fraction of a job (full gradient) in a round. As there are $\eta B \lambda$ straggling rounds faced by workers $[0 : \lambda - 1]$ in rounds $[1 : \eta(W - 1 + B)]$, the number of jobs that will be finished by non-straggling workers in these rounds is at most:

$$\lfloor \min\left(\{\eta(W - 1 + B), \eta L(n(W - 1 + B) - B\lambda)\}\right) \rfloor$$

gradients. If $L < L_B^*$, we have $\eta L(n(W - 1 + B) - B\lambda) < \eta(W - 1 + B)$ and thus, in the end of first $\eta(W - 1 + B)$ rounds, there are

$$\lceil \eta(W - 1 + B) - \eta L(n(W - 1 + B) - B\lambda) \rceil > 0$$

pending jobs among $[1 : \eta(W - 1 + B)]$ which need to be finished. The number of pending jobs clearly increases as $\eta$ increases. In order to satisfy the delay constraint $T$, it is necessary that all these pending jobs need to be processed in rounds $[\eta(W - 1 + B) + 1 : \eta(W - 1 + B) + T]$. On the other hand, the number of jobs that can be finished in these rounds is at most $\lceil TLn \rceil$ (under the best-case scenario that all the workers are non-stragglers in these $T$ rounds), which is bounded. As $T < \infty$, for a sufficiently large $\eta$, $\lceil TLn \rceil < \lceil \eta(W - 1 + B) - \eta L(n(W - 1 + B) - B\lambda) \rceil$. Thus, at least one of the jobs in $[1 : \eta(W - 1 + B)]$ cannot be finished with delay $T$ if $L < L_B^*$. Thus, we have $L \geq L_B^*$.

- $B = W$: Consider the periodic straggler pattern depicted in Fig. 9, which conforms to the $(B, W, \lambda)$-bursty straggler model when $B = W$. Here, if $L < L_B^*$, there are $\lceil \eta - \eta L(n - \lambda) \rceil$ pending jobs after rounds $[1 : \eta]$. Following similar arguments as in the $B < W$ case, for sufficiently large $\eta$ delay criterion $T$ cannot be met and thus it can be inferred that $L \geq L_B^*$.

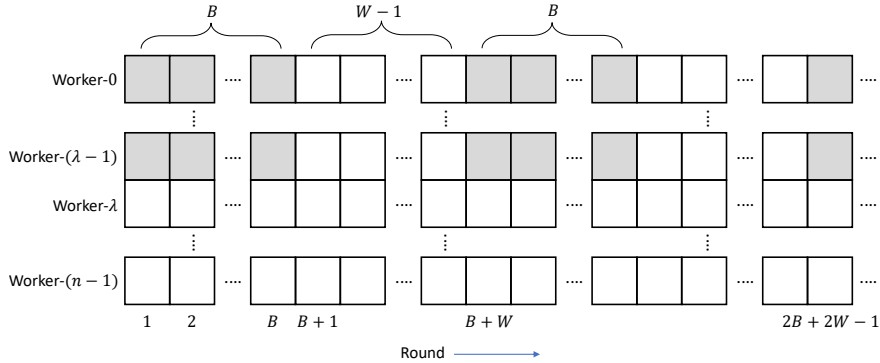

Figure 8: A periodic straggler pattern conforming to the $(B, W, \lambda)$-bursty straggler model, when $B < W$. Here, the shaded boxes indicate stragglers.

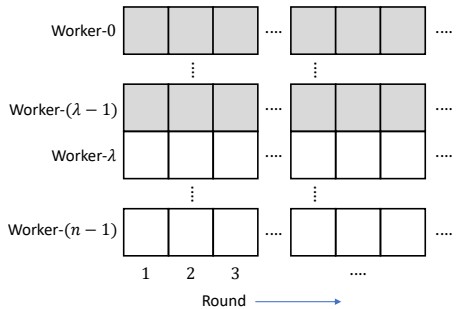

Figure 9: A periodic straggler pattern conforming to the $(B, W, \lambda)$-bursty straggler model, when $B = W$. Here, the shaded boxes indicate stragglers.

□

An analogous result can be shown with respect to $(N, W', \lambda')$-arbitrary straggler model as well.

**Theorem F.2.** *Consider any sequential gradient coding scheme, which tolerates straggler patterns permitted by the $(N, W', \lambda')$-arbitrary straggler model. Let the normalized load per worker $L$ be a constant, $T < \infty$ and the number of jobs $J \to \infty$. We have:*

$$L \geq L_A^* \triangleq \begin{cases} \frac{W'}{n(W'-N)+N(n-\lambda')}, & \text{if } N < W, \\ \frac{1}{n-\lambda}, & \text{if } N = W. \end{cases} \tag{3}$$

*Proof.* The proof here follows in a similar manner as that of Theorem F.1 and hence, details are omitted. For the case $N < W'$, consider Fig. 10 (analogous to Fig. 8). Considering first $\eta W'$ rounds, if $L < L_A^*$, number of pending jobs is given by $\lceil \eta(W') - \eta L(nW' - N\lambda) \rceil$. If $\eta$ is sufficiently large, we have $\lceil TLn \rceil < \lceil \eta W' - \eta L(nW' - N\lambda) \rceil$ and hence, at least one of the jobs cannot be finished with delay $T$. Thus $L \geq L_A^*$. When $N = W'$, if $\lambda$ is replaced with $\lambda'$ in Fig. 9, we obtain a straggler pattern conforming to the $(N, W', \lambda')$-arbitrary straggler model. The proof follows precisely as in the case of $B = W$ in Theorem F.1 (with $\lambda$ replaced by $\lambda'$).

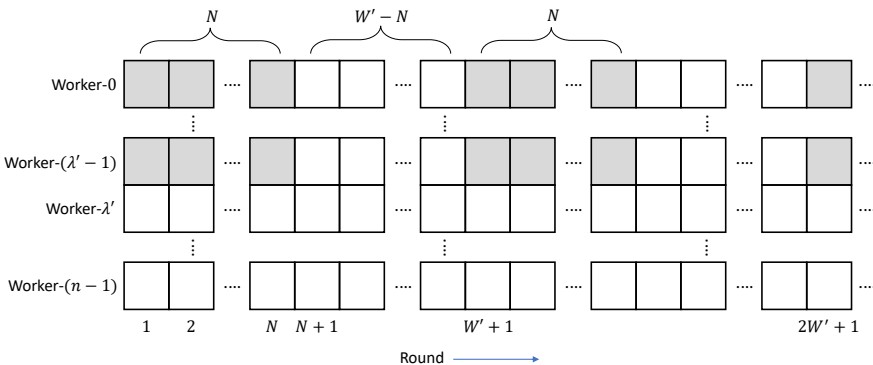

Figure 10: A periodic straggler pattern conforming to the $(N, W', \lambda')$-arbitrary straggler model, when $N < W'$.

□

**Remark F.1** (Optimality). From equation 1 and equation 2, it can be observed that when $\lambda = n-1$ or $n$, M-SGC scheme is optimal as a scheme tolerating any straggler pattern conforming to the $(B, W, \lambda)$-bursty straggler model. In an analogous manner, using equation 1 and equation 3, M-SGC can be shown to be optimal when $\lambda' = n - 1$ or $n$ (with respect to the $(N = B, W' = W + B - 1, \lambda' = \lambda)$-arbitrary straggler model). Moreover, for fixed $n$, $B$ and $\lambda$, the gap between equation 1 and equation 2 decreases as $O(\frac{1}{W})$. In Fig. 11, we plot normalized loads of SR-SGC and M-SGC schemes and compare it with equation 2. Similarly, gap between equation 1 and equation 3 decreases as $O(\frac{1}{W'})$.

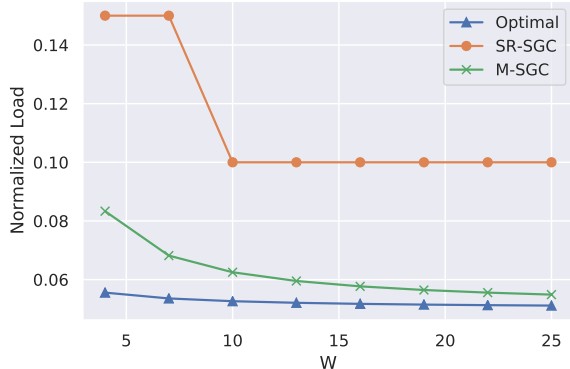

Figure 11: A comparison of normalized loads incurred by SR-SGC and M-SGC schemes when $\{n = 20, B = 3, \lambda = 4\}$ and $W$ is varied. Note however that delay parameter $T$ values in the case of SR-SGC and M-SGC schemes are given by $B$ and $W - 2 + B$, respectively. Hence, in terms of scheduling tasks (see Remark 2.1) SR-SGC scheme offers more flexibility.

**(a)**

| Worker-0 | $\ell_0(1)$ | $\ell_0(1)$ | $\ell_0(3)$ | $\ell_0(3)$ | $\ell_0(5)$ | $\ell_0(5)$ |
|---|---|---|---|---|---|---|
| Worker-1 | $\ell_1(1)$ | $\ell_1(1)$ | $\ell_1(3)$ | $\ell_1(3)$ | $\ell_1(5)$ | $\ell_1(5)$ |
| Worker-2 | $\ell_2(1)$ | $\ell_2(2)$ | $\ell_2(3)$ | $\ell_2(4)$ | $\ell_2(5)$ | $\ell_2(6)$ |
| Worker-3 | $\ell_3(1)$ | $\ell_3(2)$ | $\ell_3(3)$ | $\ell_3(4)$ | $\ell_3(5)$ | $\ell_3(6)$ |
| | 1 | 2 | 3 | 4 | 5 | 6 |

Round →

(a)

**(b)**

| Worker-0 | $g_0(1)$ / 0 | $g_0(2)$ / $g_0(1)$ | $g_0(3)$ / 0 | $g_0(4)$ / $g_0(3)$ | $g_0(5)$ / 0 | $g_0(6)$ / $g_0(5)$ |
|---|---|---|---|---|---|---|
| Worker-1 | $g_1(1)$ / 0 | $g_1(2)$ / $g_1(1)$ | $g_1(3)$ / 0 | $g_1(4)$ / $g_1(3)$ | $g_1(5)$ / 0 | $g_1(6)$ / $g_1(5)$ |
| Worker-2 | $g_2(1)$ / 0 | $g_2(2)$ / $g_2(1)$ | $g_2(3)$ / 0 | $g_2(4)$ / $g_2(3)$ | $g_2(5)$ / 0 | $g_2(6)$ / $g_2(5)$ |
| Worker-3 | $g_3(1)$ / 0 | $g_3(2)$ / $g_3(1)$ | $g_3(3)$ / 0 | $g_3(4)$ / $g_3(3)$ | $g_3(5)$ / 0 | $g_3(6)$ / $g_3(5)$ |
| | 1 | 2 | 3 | 4 | 5 | 6 |

Round →

(b)

Figure 12: Consider parameters $\{n = 4, B = 1, W = 2, \lambda = 4\}$; (a) SR-SGC scheme: reattempted tasks are indicated in red (b) M-SGC scheme.

**Example F.1.** In Fig. 12, we perform a comparison of normalized load incurred by SR-SGC and M-SGC schemes for parameters $\{n = 4, B = 1, W = 2, \lambda = 4\}$. We consider a straggler pattern conforming to the $(B = 1, W = 2, \lambda = 4)$-bursty straggler model, where in alternate rounds (rounds $1, 3, 5, \ldots$), all the workers are stragglers. In both the schemes, data set $\mathbf{D}$ is partitioned into 4 data chunks $\{\mathbf{D}_0, \mathbf{D}_1, \mathbf{D}_2, \mathbf{D}_3\}$. In Fig. 12 (a), we show the SR-SGC scheme. Here, the quantity $\ell_i(t)$ is a linear combination of 3 partial gradients $\{g_j(t)\}_{j \in [i:i+2]^*}$ (application of $(4, 2)$-GC). Hence, normalized load is $\frac{3}{4}$. On the other hand, in the case of M-SGC scheme (Fig. 12 (b)), each worker in a round computes 2 partial gradients and hence, the normalized load is $\frac{1}{2}$. In both the schemes, even though all the workers are stragglers in round-$(2t - 1)$ $(t = 1, 2, \ldots)$, master is able to compute $g(2t - 1)$ and $g(2t)$ together after receiving task results from workers in round-$2t$. In the SR-SGC scheme, the increased computational load is due to the use of $(4, 2)$-GC to compute 2 jobs in round-$2t$ (corresponding to each of jobs $(2t - 1)$ and $2t$, there are two tasks being performed in round-$2t$). Being a scheme where coding is only across workers (and not across rounds), in the $(4, 2)$-GC scheme, partial gradient computations on each data chunk are attempted by 3 distinct workers and this contributes to an increased load. On the other hand, in the M-SGC scheme, by repeating mini-tasks across rounds, we achieve optimal load which matches equation 2.

## G GRADIENT CODING: SIMPLIFICATION FOR SPECIFIC PARAMETERS

The gradient coding approach in general works for any $s$ such that $0 \leq s < n$. For instance, let $s = 2$ and $n = 6$. As per the general $(n, s)$-GC scheme (see the description in Sec. 3.1), the master partitions the dataset $\mathbf{D}$ into $n = 6$ data chunks $\mathbf{D}_0, \ldots, \mathbf{D}_5$. These data chunks are then placed across $n = 6$ workers as shown in Fig. 13. Here, worker-0 is expected to compute the partial gradients

$g_0, g_1, g_2$ and then return $\ell_0 = \alpha_{0,0}g_0 + \alpha_{0,1}g_1 + \alpha_{0,2}g_2$. Similarly, worker-1 is supposed to return $\ell_1 = \alpha_{1,1}g_1 + \alpha_{1,2}g_2 + \alpha_{1,3}g_3$ and so on. The master can reconstruct $g \triangleq g_0 + g_1 + \cdots + g_5$ as a linear combination of any $(n - s) = 4$ $\ell_i$'s. For instance, assume that workers $0, 2, 4, 5$ returned their results to master initially. The master will now perform the decoding step of computing $g = \beta_{\{0,2,4,5\},0}\ell_0 + \beta_{\{0,2,4,5\},2}\ell_2 + \beta_{\{0,2,4,5\},4}\ell_4 + \beta_{\{0,2,4,5\},5}\ell_5$. However, in the work Tandon et al. (2017), the authors note that for the special case when $(s + 1)$ divides $n$, there exists an alternative, simpler approach that only requires a trivial decoding procedure. We will now demonstrate this idea for $n = 6, s = 2$. In Fig. 14, we outline how data fragments are placed in this simplified GC scheme. Here, workers are divided into $\frac{n}{s+1} = 2$ groups. The computations done across workers within each group are simply replications of each other (for this reason, we will refer to the simplified GC scheme as GC-Rep). In particular, workers in group-0 are supposed to compute $g_0, g_1, g_2$ and return $\ell^{(0)} \triangleq g_0 + g_1 + g_2$. Similarly, workers in group-1 need to compute $g_3, g_4, g_5$ and return $\ell^{(1)} \triangleq g_3 + g_4 + g_5$. In order to decode $g$, the master just needs to compute the sum $\ell^{(0)} + \ell^{(1)}$, which is always possible if there are only $s = 2$ stragglers. In addition to the simplicity with respect to decoding, the GC-Rep scheme offers improved straggler resiliency in the following sense. If there are more than $s$ stragglers, there is no guarantee that GC will recover $g$. However, GC-Rep can tolerate the straggler pattern as long as one worker in each group is a non-straggler. For instance, the GC-Rep scheme succeeds even if workers $1, 2, 3$ and $5$ are stragglers. Since neither worker-0 nor worker-4 computes the partial gradient $g_3$, GC will fail in this scenario.

Both M-SGC and SR-SGC schemes can leverage the existence of the GC-Rep scheme. If $(\lambda + 1)$ divides $n$, we may use the GC-Rep scheme in combination with the M-SGC scheme, since the latter employs an $(n, \lambda)$-GC scheme to protect some of the computations from stragglers. We will refer to this modified M-SGC scheme as the M-SGC-Rep scheme.

Similarly, in the case of SR-SGC scheme where $s \triangleq \lceil \frac{B\lambda}{W-1+B} \rceil$, one may use GC-Rep instead of GC as the base scheme if $(s + 1)$ divides $n$ (let the new scheme be called SR-SGC-Rep). In order to exploit the fact that all workers within group-$i$ return the same result $\ell^{(i)}$ in the GC-Rep scheme, we present below a minor modification of Algorithm 1. Recall the definition of $\mathcal{N}(t)$, which denotes the number of task results corresponding to job-$t$ returned to master in round-$t$.

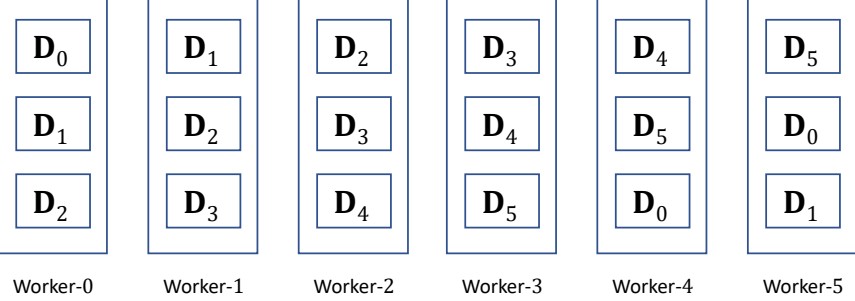

Figure 13: An illustration of the placement of data chunks across workers in the general GC scheme for parameters $n = 6, s = 2$.

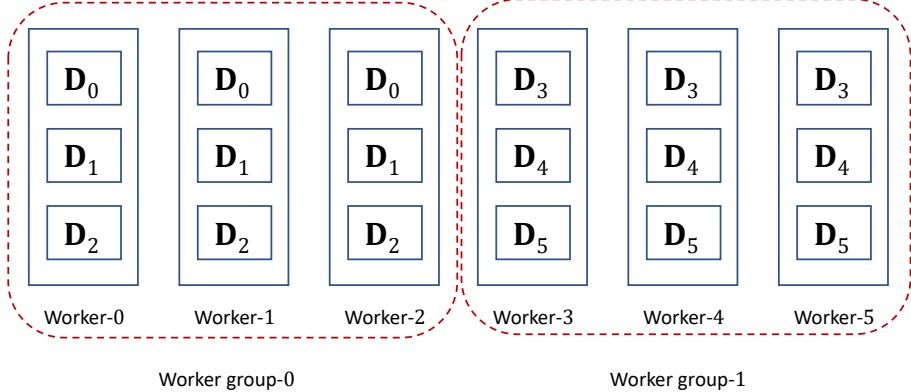

Figure 14: An illustration of the placement of data chunks across workers for the simplified GC scheme when $n = 6, s = 2$. Here, worker-$i$ belongs to group-$\lfloor \frac{i}{s+1} \rfloor$.

---

**Algorithm 3** The algorithm which may be used by the master to assign tasks in round-$t$ for the SR-SGC-Rep scheme (valid only if $(s + 1)$ divides $n$).

---

Initialize $\delta \triangleq \mathcal{N}(t - B)$.
**for** $i \in [0 : n - 1]$ **do**
    **if** $\ell^{(\lfloor \frac{i}{s+1} \rfloor)}(t - B)$ is returned by some worker in group-$\lfloor \frac{i}{s+1} \rfloor$ in round-$(t - B)$ **then**
        Worker-$i$ attempts to compute $\ell^{(\lfloor \frac{i}{s+1} \rfloor)}(t)$.
    **else if** $\delta < n - s$ **and** $\ell^{(\lfloor \frac{i}{s+1} \rfloor)}(t - B)$ is not returned previously by worker-$i$ in round-$(t - B)$
**then**
        Worker-$i$ attempts to compute $\ell^{(\lfloor \frac{i}{s+1} \rfloor)}(t - B)$.
        Set $\delta = \delta + 1$
    **else**
        Worker-$i$ attempts to compute $\ell^{(\lfloor \frac{i}{s+1} \rfloor)}(t)$.

---

It is worth noting that the computational load for GC-based and GC-Rep-based schemes is the same. Although decoding in GC-Rep-based schemes is simpler, as decoding time may be included into the master's idle time (see Appendix K), this will not have any impact on the total run time. However, enhanced straggler resilience of GC-Rep-based schemes may lessen the number of rounds in which the master must perform wait-outs, potentially reducing the overall run time.

Under identical load conditions, the SR-SGC-Rep scheme always permits a strict superset of straggler patterns compared to the GC-Rep scheme, as is the case between SR-SGC and GC. Therefore, SR-SGC-Rep is anticipated to outperform GC-Rep. A direct comparison between M-SGC-Rep and GC-Rep is not feasible in terms of the straggler patterns tolerated by these schemes. However, because M-SGC-Rep requires a substantially lower computational load than GC-Rep, it is expected to perform better than the latter.

## H    AWS LAMBDA ARCHITECTURE

This section discusses the overall architecture, limitations, and additional details about the setup and usage of the AWS cloud resources used in our experiments (Sec. 4).

We use AWS Lambda functions as workers in our distributed training experiments. Each Lambda instance has 2500 MB of RAM, 1 vCPU, and supports Python 3.8 runtime environment. A Lambda layer is used to inject external dependencies into the runtime environment (e.g. PyTorch, TorchVision, NumPy etc). Since the size of required external libraries exceeds the 200MB limit of Lambda layers, we zip some libraries in the layer package and unzip them at the time of Lambda instance creation. Note that this will not affect workers' run times as we perform a *warm-up* round before each experiment to ensure that our Lambda instances are initialized and functional.

Another limitation concerning the use of Lambda functions for training ML models is the total payload size quota of 6 MB. i.e., the total sum of payload sizes in the HTTP request and response cannot exceed 6 MB. Note that ideally the master includes current model weights in the HTTP request payload, and receives the task results via the HTTP response payload. This incurs a serious limitation on any reasonably-sized neural network. To overcome this, we need to use a proxy storage service to communicate model weights and task results.

Fortunately, we have two storage options; Amazon S3 (Simple Storage Service) and AWS EFS (Elastic File System). We use the latter, as it will provide higher throughput. EFS is a shared network file system that will be mounted on all Lambda instances at their time of creation. This way, it can be used as a means for communication between the master and workers. In our experiments, we reserve the payload limit for the task result communication, and use EFS to communicate updated model weights to workers, as depicted in Figure 15 (a). The overall architecture of our cloud resources is shown in Figure 15 (b). We use AWS Serverless Application Model (SAM) tool to define, manage, and deploy the cloud resources (included in the code submitted as supplementary material).

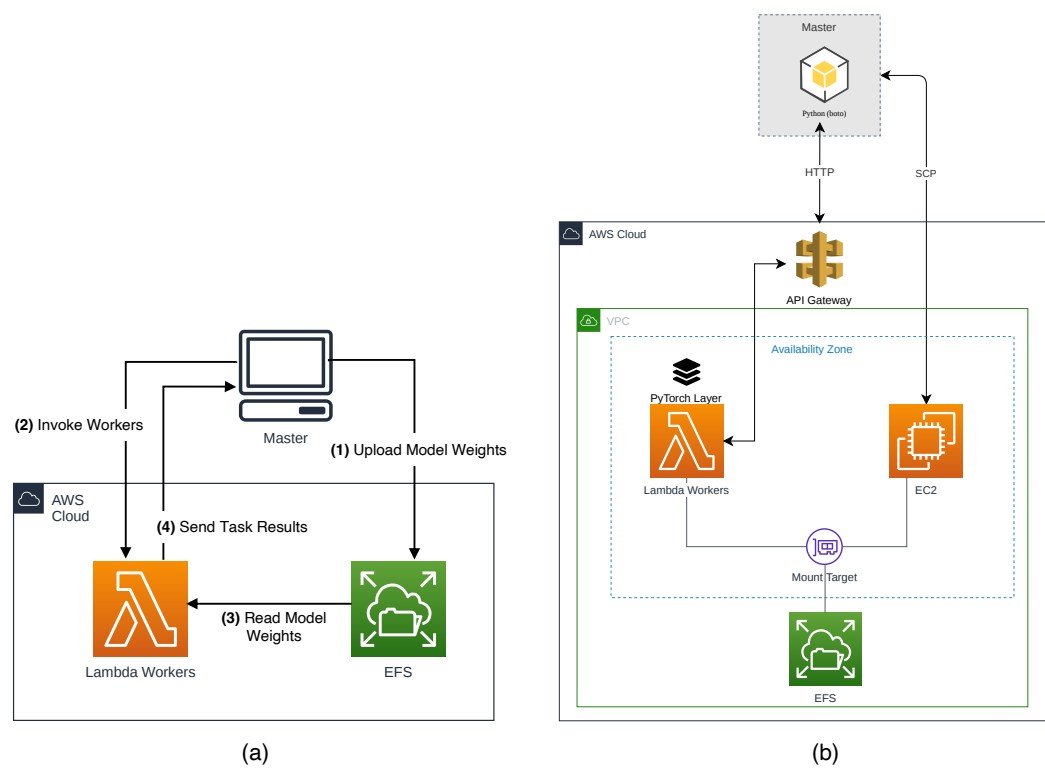

Figure 15: (a) Communication between master and Lambda workers at each round. (b) The overall architecture of AWS cloud resources used.

## I    EXPERIMENTAL SETUP AND POTENTIAL APPLICATIONS

We provide here a more detailed discussion of the experimental setup in Section 4.2. We consider training $M = 4$ neural networks, denoted by $\mathrm{NN}_1, \mathrm{NN}_2, \mathrm{NN}_3$ and $\mathrm{NN}_4$. For neural network $\mathrm{NN}_j$ where $j = 1, 2, 3, 4$, we let $\mathbf{w}_j^{(0)}$ denote the initialized weights and $\mathbf{w}_j^{(i)}$ denote the weights after a total of $i$ rounds of gradient descent updates i.e.,

$$\mathbf{w}_j^{(i)} = \mathbf{w}_j^{(i-1)} - \epsilon_j^{(i)} \mathbf{g}_j^{(i-1)} \tag{4}$$

where $\mathbf{g}_j^{(i-1)}$ denotes the gradient associated with neural network $\mathrm{NN}_j$ with weights $\mathbf{w}_j^{(i-1)}$ and $\epsilon_j^{(i)}$ denotes the learning rate. We assume that the training of $\mathrm{NN}_1, \mathrm{NN}_2, \mathrm{NN}_3$ and $\mathrm{NN}_4$ is interleaved across rounds – i.e., model updates for the *interleaved training* uses the following schedule:

- Weights of $\text{NN}_1$ are updated in rounds: 1 (Initialization), 5, 9, 13, . . .
- Weights of $\text{NN}_2$ are updated in rounds: 2 (Initialization), 6, 10, 14, . . .
- Weights of $\text{NN}_3$ are updated in rounds: 3 (Initialization), 7, 11, 15, . . .
- Weights of $\text{NN}_4$ are updated in rounds: 4 (Initialization), 8, 12, 16, . . .

Thus if we consider the training of say $\text{NN}_1$, then there two steps:

- **Step 1**: At the start of round-$(4i + 1)$, the server will generate $\mathbf{w}_1^{(i)}$. It will have decoded the gradient vector $\mathbf{g}_1^{(i-1)}$ by the end of round-$4i$ and performed the SGD update in equation 4.
- **Step 2**: The server will issue a request at the start of round-$(4i + 1)$ to the workers to compute the associated partial gradients such that the gradient vector $\mathbf{g}_1^{(i)}$ can be computed by the end of round-$(4i + 4)$.

By following a similar approach for each model it is clear that for neural network $\text{NN}_j$, when computing the gradient $\mathbf{g}_j^{(i)}$, a request will be issued by the server to the clients at the start of round-$(4i + j)$ and the computation must be completed by the end of round-$(4i + j + 3)$. Thus our interleaved approach is designed so that that each of the gradient vectors can be decoded within a span of $M = 4$ rounds.

Our method can be viewed as a pipelined approach for training of multiple neural networks. Note that in our current setting we assume that the parameters of only one model can be updated in each round, which arises in resource constrained devices. Our method also naturally complements other approaches for parallel training of multiple models and leverages on the temporal structure of straggler patterns to achieve speedups. We also emphasize that we do not require the multiple neural network models be trained on the same dataset. Each neural network model can be trained on a different dataset. We also do not require that the architecture of the neural network models being trained to be identical. Nevertheless we point out that our setting would be most efficient when the compute time for the gradients for each model is approximately the same. We believe this is a rather benign requirement. Finally we discuss a few applications where multiple-model training arise naturally.

- In the training of deep learning models, we are often required to perform a search over various hyperparameters and this is done through some form of a grid search (Bergstra et al., 2011). Each choice of hyperparameter corresponds to a new model. Ultimately the ideal hyperparameters are selected through a validation set.
- In ensemble learning (Zhou, 2012), a number of models need to be trained simultaneously. Their predictions are then combined through some averaging mechanism.
- Since our approach can use completely different datasets for each of the models, it is also applicable in settings of "multi-model learning" (Bhuyan et al., 2022; da Silva et al., 2022), where multiple datasets are used for different models. For instance, sensors deployed in time-varying or periodic environment (e.g., day/night camera images, orbiting satellite data etc.) or collecting different modalities (speech, images etc.) would naturally generate multiple datasets and different models should be trained for each one. Multi-model learning has also been applied in real-time video surveillance applications (Wu et al., 2021).

## J    SELECTING CODING SCHEME PARAMETERS

This section discusses the parameter selection method used for our proposed sequential gradient coding schemes, SR-SGC and M-SGC. We begin by noting that the total number of valid parameter combinations for each of these schemes are too large for a grid search to be feasible, as evaluation of each parameter combination requires training models for multiple rounds. Instead, we utilize the observation that increasing the normalized load will linearly increase the average runtime of the workers. Fig. 16 shows the average job completion time of 256 workers across 100 rounds for multiple values of load in $[0, 1]$.

We can exploit the observation above to estimate the delay profile corresponding to various coding schemes with variable normalized loads. After estimating the slope of linear fit in Fig. 16 (let this

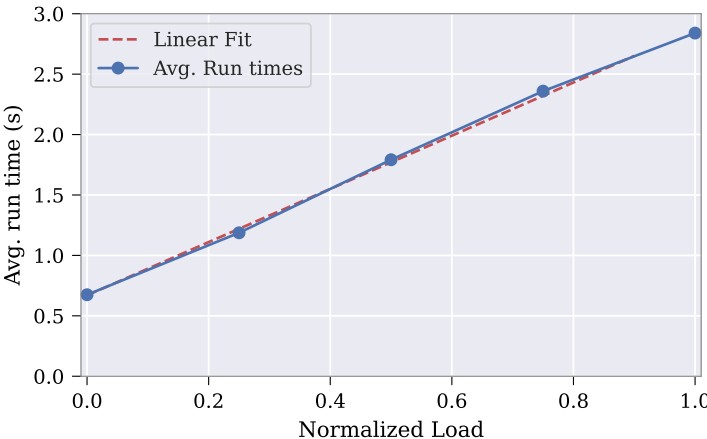

Figure 16: Average run time (256 workers, 100 rounds) scales linearly with workers' computational load.

be $\alpha$), we run our distributed training experiment for $T_{\text{probe}} = 80$ jobs with no coding, and store the observed runtime of workers across rounds (we call this the *reference delay profile*). Note that the normalized worker load in case of no coding will be $1/n$.

Next, consider a coding scheme with fixed parameters $B, W, \lambda$, and a fixed normalized load of $L$. We can estimate the runtime of our coding scheme by feeding the reference delay profile to the master node to simulate the run time of workers. Based on our previous discussion, we should take into account the increase in the workers' run time due to the increase of the load from $1/n$ to $L$, by adding $(L - 1/n)\alpha$ seconds to our reference delay profile. Using this *load-adjusted* delay profile and the considered coding scheme, the master node will try to resolve stragglers at each round, and if not, it will wait out all the workers, resulting in a simulated total run time for the coding scheme.

Fig. 17 shows the estimated runtime of $80$ rounds for different choices of parameters $B, W, \lambda$ in SR-SGC and M-SGC. For each of the two schemes, we select the parameters corresponding to the smallest estimated training runtime, denoted by the blue circles on Fig. 17 and listed in Table 1.

### J.1 SENSITIVITY TO PARAMETERS

The Fig. 17 also demonstrates sensitivity of the coding schemes to their parameters, where we can observe smooth changes in training time with respect to coding parameters.

For SR-SGC, the normalized load varies significantly with the choice of parameters: $L = \frac{s+1}{n}$ with $s = \lceil \frac{B\lambda}{W-1+B} \rceil$. Specifically, the load is directly proportional to $\lambda$. As observed in Fig. 17 (left), for each choice of $B$ and $W$, increasing $\lambda$ above a certain threshold leads to a significant increase in the runtime. Therefore, $\lambda$ should be chosen carefully as it will affect the load and runtime heavily.

On the other hand, the computational load in M-SGC is less dependent on selected parameters, as the load is upper-bounded by $2/n$ (see Remark 3.3). Therefore, the choice of $\lambda$ does not play a crucial role as long as it is above the typical number of stragglers. Also, as observed in Fig. 17 (right), runtime is fairly insensitive to the choice of $B$ as long as $W$ and $B$ are close.

**Remark J.1.** Recommendations for selecting parameters of M-SGC and SR-SGC:

- Keeping $W$ close to $B$ seems to be the right rule of thumb for both schemes. Also, the dependence of both schemes on $B$ is less critical and increasing $W$ is generally not preferred as it reduces the straggler correction capability of the coding schemes.

- For M-SGC, the choice of $\lambda$ is not critical as long as it is above a certain threshold, but for SR-SGC it is an important consideration as it affects the load significantly.

- Therefore, it is recommended to start with a fixed $B$, choose $W$ as close as possible to $B$, and find a large enough $\lambda$ for M-SGC or a small enough $\lambda$ for SR-SGC, based on the straggler pattern.

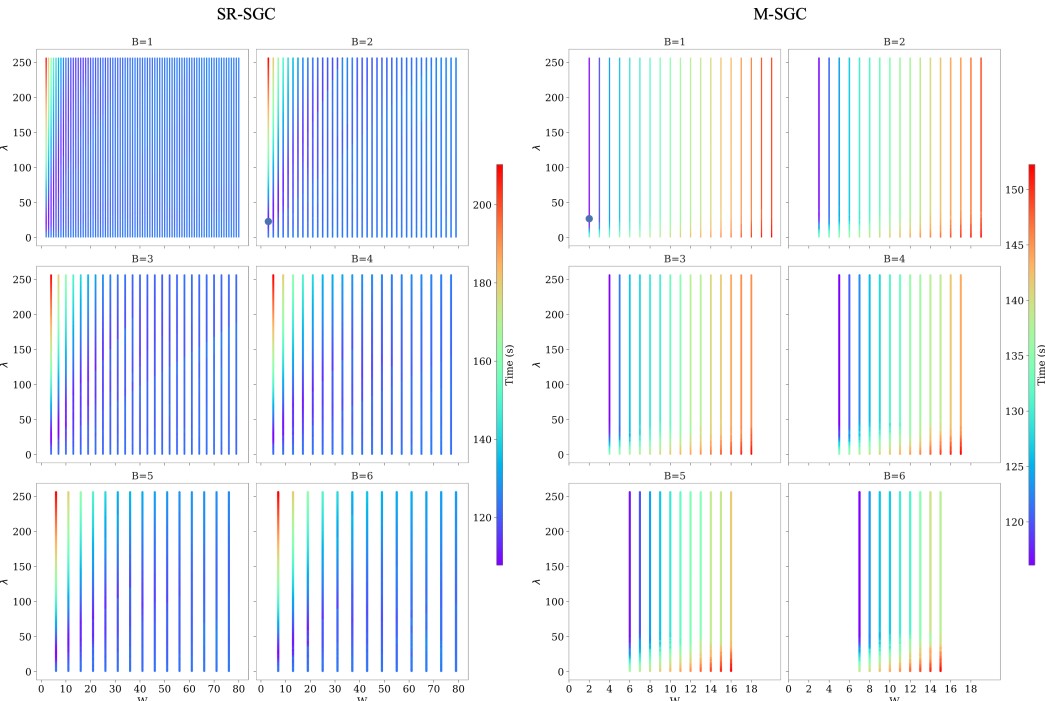

Figure 17: Estimated runtime of 80 rounds for different choices of parameters $B, W, \lambda$. Left: SR-SGC, Right: M-SGC. Blue dot marks the shortest runtime (selected parameters).

Next, we evaluate the sensitivity of parameter selection to the number of rounds used in the selection process $T_{\text{probe}}$. Table 3 lists the selected parameters using different values of $T_{\text{probe}}$. For each unique set of parameters, we performed the experiments discussed in Section 4.2 10 times and included the average runtime as well. As expected, we observe a general trend of improvement in total training time by increasing $T_{\text{probe}}$. Moreover, for M-SGC even few number of rounds are enough to tune the coding scheme, as M-SGC with parameters selected over only 10 rounds outperforms other coding schemes across all other values of $T_{\text{probe}}$.

Table 3: Selected parameters using different values of $T_{\text{probe}}$.

| Scheme | $T_{\text{probe}}$ | Selected Parameter | Load | Runtime (avg. $\pm$ std.) |
|---|---|---|---|---|
| | 10 | (1, 2, 24) | 0.007512 | 872.78 ± 80.67 (s) |
| | 20 | (1, 2, 24) | 0.007512 | 872.78 ± 80.67 (s) |
| M-SGC $(B, W, \lambda)$ | 40 | (1, 2, 27) | 0.007543 | 871.99 ± 59.76 (s) |
| | 60 | (1, 2, 27) | 0.007543 | 871.99 ± 59.76 (s) |
| | 80 | (1, 2, 27) | 0.007543 | 871.99 ± 59.76 (s) |
| | 10 | (2, 3, 15) | 0.035156 | 1226.90 ± 93.53 (s) |
| | 20 | (2, 3, 15) | 0.035156 | 1226.90 ± 93.53 (s) |
| SR-SGC $(B, W, \lambda)$ | 40 | (2, 3, 20) | 0.042969 | 1060.61 ± 62.72 (s) |
| | 60 | (2, 3, 22) | 0.046875 | 964.06 ± 53.48 (s) |
| | 80 | (2, 3, 23) | 0.050781 | 984.37 ± 51.02 (s) |
| | 10 | (9) | 0.039062 | 1288.57 ± 94.15 (s) |
| | 20 | (10) | 0.042969 | 1261.61 ± 80.87 (s) |
| GC $(s)$ | 40 | (11) | 0.046875 | 1168.67 ± 77.60 (s) |
| | 60 | (14) | 0.058594 | 1142.01 ± 38.41 (s) |
| | 80 | (15) | 0.062500 | 1067.56 ± 40.92 (s) |

## K    ANALYSIS OF OVERHEADS

In this section we discuss two overheads in proposed sequential gradient coding schemes, as well as ways to effectively remove those overheads in practice.

### K.1    DECODING TIME

At the end of each round, the master node decodes the task results obtained from a subset of workers to calculate gradients. Both sequential coding methods proposed in this paper use GC as their core coding/decoding method: SR-SGC with parameters $\{B, W, \lambda\}$ uses an $(n, \lceil \frac{B\lambda}{W-1+B} \rceil)$-GC, and M-SGC uses $B$ independent $(n, \lambda)$-GC schemes. In $(n, s)$-GC, gradients are simply obtained by a linear combination of task results from $n - s$ workers (Tandon et al., 2017). Therefore, decoding consists of two steps:

1. Finding the decoding coefficients based on the observed straggler pattern. This can be done by solving a linear matrix equation.

2. Calculating the linear combination of task results from non-straggling workers, using the coefficients from step 1.

Table 4 summarizes the decoding time for the main experiment presented in Section 4.2.

Table 4: Decoding time of different schemes

| Scheme | Parameters | Decoding Time (avg. $\pm$ std.) | Longest Decoding | Fastest Round |
|---|---|---|---|---|
| M-SGC | $B = 1, W = 2, \lambda = 27$ | 290 ± 18.7 ms | 479 ms | 1177 ms |
| SR-SGC | $B = 2, W = 3, \lambda = 23$ | 309 ± 20.1 ms | 401 ms | 1349 ms |
| GC | $s = 15$ | 204 ± 20.6 ms | 408 ms | 1379 ms |

We would like to point out that if training of more than $T + 1$ models is pipelined, the decoding can be done in the master's idle time, and the decoding time will not add any overhead to the training time. As stated in Remark 2.1 and further explained in Appendix I, when the decoding delay of the sequential coding scheme is $T$ rounds, training at least $M = T + 1$ models needs to be pipelined. Specifically, with $M$ models and decoding delay of $T$ rounds, the gradient of each model is ready to be decoded $D = M - T - 1$ rounds earlier than the next round of the model begins. In case of $M = T + 1$, the gradients of a model are ready for decoding just before the next round for the model begins ($D = 0$). However, when $M > T + 1$, i.e. $D > 0$, the master node has at least one

gap round to decode the gradients, and therefore can run the decoding at its idle time while workers are performing their assigned tasks. We emphasize that this is indeed applicable to the experiments presented in Section 4.2, where $M = 4$ and the delay of all models are smaller than $M - 1 = 3$.

As an example, let us consider the SR-SGC scheme with parameters $\{B = 2, W = 3, \lambda = 23\}$ discussed in Section 4.2, where $M = 4$ and $T = 2$. Calculations of gradients for model 1 start at rounds $t_1 = 1, t_2 = 5, t_3 = 9, \cdots$. Calculating the first gradients of model 1 starts at round $t_1 = 1$. By the end of round $t_1 + T = 3$, the gradients are ready to be decoded. Note that the second gradients of model 1 are to be calculated at round $t_2 = 5$. Therefore, the master node can perform the decoding during its idle time over round 4, when it is waiting for the worker nodes (note that as can be seen from Table 4, the longest decoding time is shorter than the fastest round time). This way, the gradients are decoded before round $t_2 = 5$ begins and thus no decoding overhead is imposed.

### K.2 PARAMETER SELECTION TIME

In Section J, we discussed the process of selecting parameters $\{B, W, \lambda\}$ for the coding schemes. This process requires running uncoded training for some number of rounds ($T_{\text{probe}}$) and storing the job completion time of workers. This delay profile is then used to search for the best coding parameters. In the experiment section 4.2, we started coded training from round-1 (delay profile measurement and selection of best parameters are done beforehand). However, in practice, one can opt to start with uncoded training in round-1 and then switch to coded training after $T_{\text{probe}}$ uncoded rounds (a few additional rounds will be needed to perform the exhaustive search for the best parameters as well). This way, the time to be spent initially for delay profile measurements can be utilized towards completion of jobs.

Fig. 18 demonstrates this method in which training starts uncoded, and after $T_{\text{probe}} = 40$ rounds, master node uses the observed delay profile from the past rounds to perform an exhaustive search for the best parameters of the coding schemes. In this experiment, it took $\sim 8$ seconds for SR-SGC, $\sim 2$ seconds for M-SGC, and less than a second for GC to search over all valid parameters. The training is then switched to coded mode after the search is over. Here, we still observe significant gains from M-SGC compared to uncoded and GC methods.

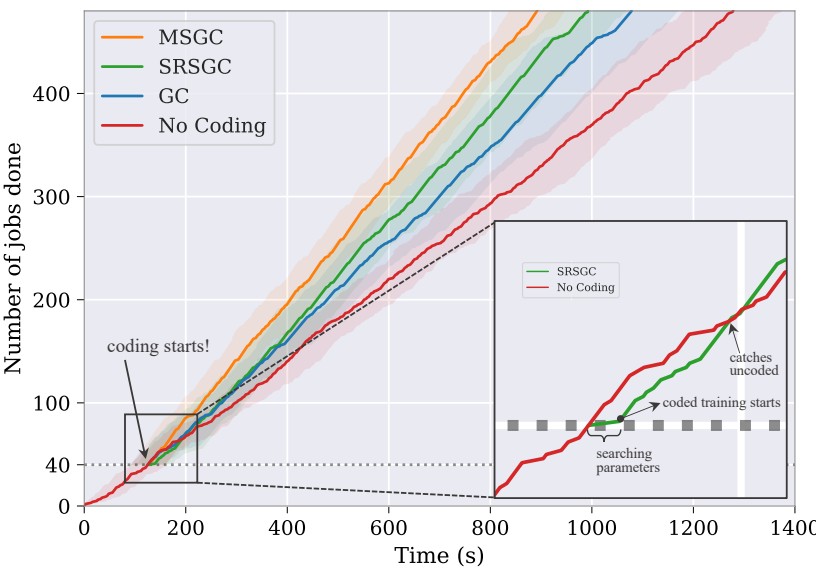

Figure 18: The same setup as in Section 4.2, but training starts uncoded and switches to coded mode after $T_{\text{probe}} = 40$ rounds. The delay profile measured from the initial 40 rounds is used to select the coding parameters. Plot shows average $\pm$ std. over 10 independent trials. Transition to SR-SGC is zoomed in.

## L    TRAINING RESNET-18 ON CIFAR-100

Here, we present the results of concurrently training $M = 4$ ResNet-18 models on CIFAR-100 over
AWS Lambda, using the different coding schemes. ResNet-18 has 11,227,812 parameters and hence,
the size of model weights and gradient updates are roughly 22.5 MB in 16-bit floating point format.
This is much larger than the 6MB payload size limit of AWS Lambda (see Section H). This means
the master node and workers have to essentially use a shared storage (Amazon EFS) to communicate
the model weights and coded gradients at each round, as depicted in Fig. 19 (a). At each round, the
master node first uploads updated model weights to the dedicated shared storage and invokes Lambda
workers via an HTTP request. Lambda workers then read the model weights from the storage and
proceed to calculate the task results. Finally, each worker uploads the coded gradients to the shared
storage and signals the master node via the HTTP response.

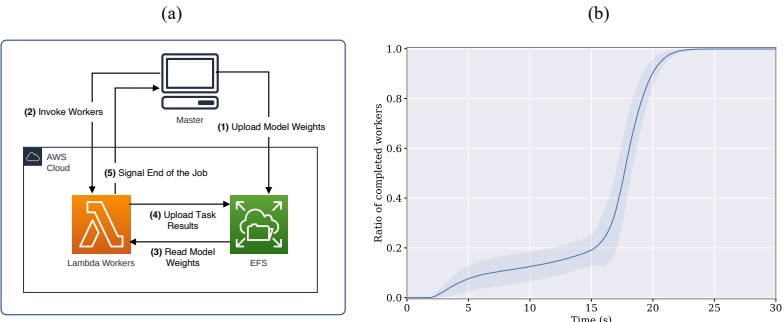

Figure 19: (a) Communication between master and Lambda workers at each round. (b) Empirical
CDF of workers' completion time, averaged over 100 rounds and 256 workers (shades represent
standard deviation. Each worker calculates gradients for a batch of 2 CIFAR-100 images on ResNet-
18, and uploads the gradients to EFS.

The process of uploading the task results to the shared storage will significantly increase the commu-
nication delay, given the write throughput limits of EFS. This is clearly observed in the empirical
CDF of workers' completion time in Fig. 19 (b). Note that for the CNN model used in Section 4, the
task results were directly sent to the master node in the payload of the HTTP response. Therefore,
given the increased variance of workers' completion time, we opt to choose a higher value of $\mu = 5$
for this case to let more workers finish at each round.

We used $n = 256$ Lambda workers to train $M = 4$ models concurrently for 1000 rounds (250 rounds
for each classifier). A batch size of 512 samples and ADAM optimizer is used. Fig. 20 plots the the
number of completed jobs (for all $M = 4$ models) over time for the three coding schemes, as well as
uncoded training. Coding parameters are selected based on the method discussed in Section J using
$T_{\text{probe}} = 20$ rounds. The results showed that M-SGC finishes training 11.6% faster than GC (while
maintaining a significantly smaller normalized load), and 21.5% faster than uncoded training.

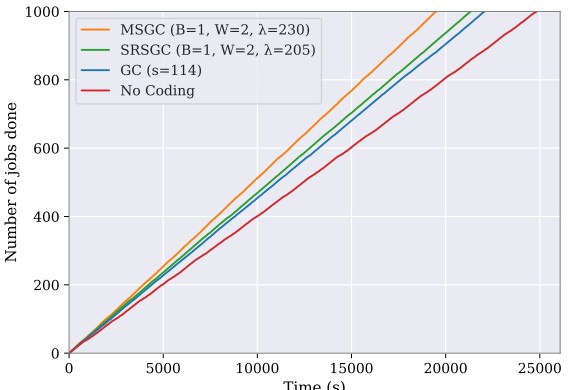

Figure 20: Training 4 ResNet-18 models on CIFAR-100. Number of completed rounds (for all
$M = 4$ models) vs. time.

