# OpenReview forum: "Sequential Gradient Coding For Straggler Mitigation"
_ICLR.cc/2023/Conference — ICLR 2023 poster_

### Official Review · Reviewer_euSx · 2022-10-25

**Confidence:** 4
**Correctness:** 4
**Technical Novelty And Significance:** 2
**Empirical Novelty And Significance:** 3
**Recommendation:** 8

**Clarity, Quality, Novelty And Reproducibility:**

The high level ideas and goals are clear, and explained well, but the notation and details are rather cumbersome.

**Strength And Weaknesses:**

Strength:
Using the temporal dimension in GC improves the normalized load per worker. The schemes cover various straggler models (e.g., bursty, arbitrary, fixed number per round). The experimental evaluation is performed in a real AWS environment, with a sizable number of nodes.

Weaknesses:
The presentation is often hard to follow, due to the heavy and involved notation, although the high level ideas are relatively clear.

**Summary Of The Paper:**

Gradient Coding is a general approach distributed computation (of gradients) in the presence of straggling nodes. The paper introduces two new schemes for GC, where coding is not only done across compute nodes, but also across the time dimension, measured by compute rounds. The first scheme uses the original GC but with selective repetition of unfinished tasks. The second scheme, and main contribution, combines two classes of tasks, one that uses GC and the other that uses repetition. The two classes are multiplexed over workers and rounds in an adaptive manner.  An experimental evaluation is performed on AWS with 256 worker nodes, showing the performance of the new scheme.

**Summary Of The Review:**

The paper proposes new CG schemes where that use the temporal dimension, and coding is done both across worker nodes and across compute rounds. As a result the normalized load per worker is reduced. Various design parameters are used to accommodate for different straggler models. The experimental evaluation is very good, and shows the performance of the new scheme.

---

> ### Author Response · Authors · 2022-11-19
> **Response to Reviewer euSx**
>
> We appreciate the reviewer's very positive assessment of our paper. A table of notations has now been included (Appendix A). We anticipate that it will help the readers and improve the paper's general readability.
>
> We would also like to point out that we have made significant additions/revisions to the manuscript including discussion on overheads and sensitivity to parameters, additional experiment of training ResNet-18 models on CIFAR-100, and discussions on the applicability of our schemes to various practical use cases.

---

### Official Review · Reviewer_dEGh · 2022-10-26

**Confidence:** 3
**Correctness:** 4
**Technical Novelty And Significance:** 3
**Empirical Novelty And Significance:** 3
**Recommendation:** 6

**Clarity, Quality, Novelty And Reproducibility:**

This paper is well written with enough details provided for reproducing the experiments.
The proposed algorithms are novel.

**Strength And Weaknesses:**

Strength:
1. This paper propose 2 extensions of gradient coding, by adding extra tolerance of stragglers and introducing asynchrony (staleness) to the updates. The idea is natural and makes sense.
2. The theoretical analysis is provided for the computational load.
3. The empirical results show that the proposed schemes are more efficient than the previous works.

Weakness:
1. The experiment setup (CNN + MNIST) is too simple and small for distributed training. Such a simple task can be finished quickly on a single computer. To justify the efficiency of the proposed methods in practice, larger tasks such as resnet on cifar-100 or imagenet is highly recommended.
2. I recommend to show the sensitivity to the hyperparameters, i.e., how the curve of convergence (training loss) vs. time changes when using different B and W. It is important to show the readers how robust the proposed algorithms are, which will also be helpful for the new users when tuning these hyperparameters.
3. It seems that no convergence proof is provided. Although I think the convergence proof will be trivial since the proof can be reduced to the one for a simple asynchronous SGD algorithm, a convergence proof will make this work more thorough.

**Summary Of The Paper:**

This paper propose 2 extensions of gradient coding, by adding extra tolerance of stragglers and introducing asynchrony (staleness) to the updates. The theoretical analysis is provided for the computational load. The empirical results show that the proposed schemes are more efficient than the previous works.

**Summary Of The Review:**

This paper propose 2 extensions of gradient coding, by adding extra tolerance of stragglers and introducing asynchrony (staleness) to the updates. The theoretical analysis is provided for the computational load. The empirical results show that the proposed schemes are more efficient than the previous works. I recommend to add experiments for the algorithms' sensitivity to the hyperparameters, and theoretical analysis for the convergence.

---

> ### Author Response · Authors · 2022-11-19
> **Response to Reviewer dEGh (part 1/2)**
>
> We thank the reviewer for their careful reading and constructive feedback towards improving our manuscript. We address the reviewer’s points below and have updated our manuscript accordingly.
>
> > “To justify the efficiency of the proposed methods in practice, larger tasks such as resnet on cifar-100 or imagenet is highly recommended.”
>
> We performed the experiment suggested by the reviewer and included the results in Appendix L.  We would like to point out that given the larger size of ResNet-18, workers have to use a different communication method to return the task results back to the master node, effectively increasing the communication time. Nonetheless, the coding schemes compare similarly to the experiment on the CNN model. We would like to refer the reviewer to Appendix L for the results.
>
>
> > “I recommend to show the sensitivity to the hyperparameters, i.e., how the curve of convergence (training loss) vs. time changes when using different B and W. It is important to show the readers how robust the proposed algorithms are, which will also be helpful for the new users when tuning these hyperparameters”
>
>
> We thank the reviewer for the helpful feedback. Appendix Section J.1 now includes detailed discussions and experiment results on sensitivity to coding parameters.
>
> First, we evaluate the sensitivity of parameters selection to the number of rounds used in the selection process, $T_\text{probe}$. Table 3 lists the selected parameters using different values of $T_\text{probe}$. For each unique set of parameters, we performed the experiments discussed in Section 4.2 ten times and included the average runtime. As expected, we observe a general trend of improvement in total training time by increasing $T_\text{probe}$. Moreover, for M-SGC even a few rounds are enough to tune the coding scheme, as M-SGC with parameters selected over only 10 rounds outperforms other coding schemes across a wide range of $T_\text{probe}$ values.
>
> We also compare the (estimated) runtime of coded training over a wide range of parameters in Figure 17.   For SR-SGC, the normalized load varies significantly with the choice of parameters. Specifically, the load is directly proportional to $\lambda$. As observed in Figure17 (left), for each choice of $B$ and $W$, increasing $\lambda$ above a certain threshold leads to a significant increase in the runtime. Therefore, $\lambda$ should be chosen carefully as it will affect the load and runtime heavily.
>
> On the other hand, the computational load in M-SGC is less dependent on selected parameters, as the load is upper-bounded by $2/n$ (see Remark 3.3). Therefore, the choice of $\lambda$ does not play a crucial role as long as it is above the typical number of stragglers. Also, as observed in Figure 17 (right), runtime is fairly insensitive to the choice of $B$ as long as $W$ and $B$ are close.
>
> To conclude, the following are our recommendations for selecting parameters of M-SGC and SR-SGC:
>
> - Keeping $W$ close to $B$ seems to be the right rule of thumb for both schemes. Also, the dependence of both schemes on $B$ is less critical and increasing $W$ is generally not preferred as it reduces the straggler correction capability of the coding scheme.
>
> - For M-SGC the choice of $\lambda$ is not critical as long as it is above a certain threshold, but for SR-SGC it is an important consideration as it affects the load significantly.
>
> - Therefore, it is recommended to start with a fixed $B$, choose $W$ as close as possible to $B$, and find a large enough $\lambda$ for M-SGC or a small enough $\lambda$ for SR-SGC, based on the straggler pattern.

---

> > ### Author Response · Authors · 2022-11-19
> > **Response to Reviewer dEGh (part 2/2)**
> >
> > > “It seems that no convergence proof is provided. Although I think the convergence proof will be trivial since the proof can be reduced to the one for a simple asynchronous SGD algorithm, a convergence proof will make this work more thorough”
> >
> >
> > First, we would like to clarify that although it is not necessary for the gradient computation of each round to be immediately available in the next round, we do not have to use stale gradients in the update of model parameters. This is because we interleave the training of different neural network models as clarified below.
> >
> > We consider training $M=4$ neural networks, denoted by $NN_1$, $NN_2$, $NN_3$, and $NN_4$ (as we did in the experimental section). For neural network $NN_j$ (where $j=1,2,3,4$), we let ${\bf w}_j^{(0)}$ denote the initial weights and ${\bf w}_j^{(i)}$ denote the weights after a total of $i$ rounds of gradient descent updates i.e.,
> >
> > ${\bf w}_j^{(i)} = {\bf w}_j^{(i-1)} - \epsilon_j^{(i)} {\bf g}_j^{(i-1)}$
> >
> > where ${\bf g}_j^{(i-1)}$ denotes the gradient associated with neural network $NN_j$ with weights $\bf w_j^{(i-1)}$ and $\epsilon_j^{(i)}$ denotes the learning rate.
> >
> > We assume that the training of $NN_1$, $NN_2$, $NN_3$, and $NN_4$ is interleaved across rounds. In particular, the model updates for the {\em interleaved training} uses the following schedule:
> > - Weights of $NN_1$ are updated in the following rounds: 1 (Initialization), 5, 9, 13 ..
> > - Weights of $NN_2$ are updated in the following rounds: 2 (Initialization), 6, 10, 14,…
> > - Weights of $NN_3$ are updated in the following rounds: 3 (Initialization), 7, 11, 15,…
> > - Weights of $NN_4$ are updated in the following rounds: 4 (Initialization), 8, 12, 16,…
> >
> > Thus if we consider the training of one such neural network, say $NN_1$, then there are the following steps:
> >
> > - **Step 1**: At the start of round $4i+ 1$, the server will generate ${\bf w}_1^{(i)}$. It will have decoded the gradient vector ${\bf g}_1^{(i-1)}$ by the end of round $4i$ and performed the SGD update noted above.
> > - **Step 2**: The server will issue a request at the start of round $4i +1$ to the client nodes to compute the associated partial gradients such that the gradient vector ${\bf{g}_1^{(i)}}$ can be computed by the end of round $4 i + 4$.
> >
> > By following a similar approach for each model it is clear that for neural network $NN_j$, when computing the gradient ${\bf g}_j^{(i)}$, a request will be issued by the server to the clients at the start of round $4i+j$ and the computation must be completed by the end of round $4i+j +3$. Thus our interleaved approach is designed so that each of the gradient vectors can be decoded within a span of $M=4$ rounds.
> >
> > We have added a clarification of the training process along these lines in Appendix I. We hope that the reviewer is convinced that we do not need to rely on the stale gradients in the model updates. For training $M$ models, with $r$ rounds per model, we need a total of $M*r$ rounds. Our accuracy will be the same as in standard (synchronous) gradient descent as a function for the same number of rounds. On the other hand, by interleaving the training of multiple models, we reduce the wait-time per round and thereby are able to reduce the total training time.

---

### Official Review · Reviewer_9BzY · 2022-10-28

**Confidence:** 4
**Correctness:** 4
**Technical Novelty And Significance:** 3
**Empirical Novelty And Significance:** 2
**Recommendation:** 6

**Clarity, Quality, Novelty And Reproducibility:**

In Example 3.3.1, the authors claim that "if the master did not receive $g_{2i}(t)$ in round $t$ and round $t+2, ...$.
However, as per my understanding of the model, this can never happen for that example. If $g_{2i}(t)$ does not return, it is because the node straggles in iteration $t$. If it straggles in iteration $t$, it then returns that answer in round $t+B = t+2$, and as per the coding scheme, $g_{2i}(t)$ is necessarily repeated in round $t+2$. Please correct this or add clarifying explanations.

Apart from the above detail, the system model and coding scheme (which is quite intricate) is presented quite clearly.

--> A small clarification in the explanation of Fig. 14: For a given normalized load, are there 80 "dots" on the y axis corresponding to each job? If this is the case, the figure is the result of several hundreds of jobs?
If not, then it is unclear what the dots represent.

**Strength And Weaknesses:**

Strengths:
+The redundancy of gradient coding can be quite a lot, and the recognition of structure among the set of stragglers in AWS lambda and reducing the overall redundancy is a positive contribution.
+The developed coding schemes are novel and clever - even though they do build upon the idea of gradient coding.
+ The results are experimentally validated, and the experimental results are quite promising, and indicate the usefulness of carefully designed redundancy in AWS Lambda.

Weaknesses:
- The main use case for the developed technique is the application of training $M$ distinct models on the same data set. This seems to be a narrow - perhaps even contrived - use case as I see it. It is also inevitable that the developed technique cannot be applied to the most common use case of training a data set, due to the coding scheme incurring a delay in decoding (the sum of partial gradients at iteration $t$ is decoded at iteration $t+B$).

- In gradient coding, the load is $s+1$ times the load of the common case. Note that this is the same overhead as if the data sets were repeated $s$ times (which is much simpler!). The main benefit of gradient coding, as I understand it, is that it does not necessarily require $s+1$ to divide $n$, whereas replication would require this. In fact, notice that this replication scheme is also a candidate for the experimental results in Fig. 4, as $s=15,n=256$.

I would like the authors to clarify if the above property is also true for the developed coding schemes. If we allowed some additional divisibility criteria to be satisfied among the parameters, then do simpler schemes such as replication match the performance of the developed schemes?

- There is no statement/result on whether the developed schemes can be improved, or gap to optimality.





**Summary Of The Paper:**

The paper develops schemes to protect distributed gradient descent from stragglers building on previous work that induces redundancy in computation via error correcting codes. The authors specifically build on a technique called gradient coding that enables a master node to recover the sum of the gradients from $n$ distributed works, even if a certain prescribed number ($s$) nodes straggle (i.e., fail to provide their computation within a time-out).

While gradient coding can protect each of the $\binom{n}{s}$ straggling patterns in every iteration, the authors here aim to be less conservative and aim to protect a subset of the straggling patterns that seem to occur more commonly in practice. They make two assumptions (1) there is some negative correlation in the straggling patterns - a node that  straggles in a given iteration is assumed to not straggle in $B$ iterations later, and (2) that the number of distinct stragglers in any given time-window is limited. For these assumptions, the authors develop new coding schemes that improve upon gradient coding in terms of the amount of redundancy required.

**Summary Of The Review:**

The paper advances the development of introducing coding based redundancy in training towards mitigating stragglers, and builds on a recent line of work that is of interest to the machine learning community. The recognition of spatial and temporal dependencies in stragglers, and the corresponding coding techniques are novel. However, the application of the techniques is limited to a relatively narrow use case of concurrent training of multiple models on the same data set.

---

> ### Author Response · Authors · 2022-11-19
> **Response to Reviewer 9BzY (part 1/2)**
>
> We thank the reviewer for their detailed review of our manuscript and helpful feedback. We address the reviewer’s points below and have updated our manuscript accordingly.
>
> > “The main use case for the developed technique is the application of training M distinct models on the same data set. This seems to be a narrow use case. It is also inevitable that the developed technique cannot be applied to the most common use case of training a data set.”
>
> We would like to clarify that in our setting we **do not** require that 1) the distinct models be trained on the same dataset or 2) the architecture of the models to be trained need not be identical. We had already noted this in Section II of the original submission and have further clarified it in the revised version of the paper. Nevertheless, we point out that our setting would be most efficient when the compute time for the gradients for each model is approximately the same. We believe this is a rather benign requirement. Given the generality of our approach, there are a number of applications where our setting applies. We list a few below:
>
> 1. In the training of deep learning models, we are often required to perform a search over various hyperparameters and this is done through some form of a grid search [R1]. Each choice of hyperparameter corresponds to a new model. Ultimately the ideal hyperparameters are selected through a validation set.
>
> 2. In ensemble learning [R2], a number of models need to be trained simultaneously. Their predictions are then combined through some averaging mechanism.
>
> 3. Since our approach can use completely different datasets for each of the models, it is also applicable in settings of “multi-model learning” [R3, R4], where multiple datasets are used for different models. For instance, sensors deployed in time-varying or periodic environment (e.g. day/night camera images, orbiting satellite data etc) or collecting different modalities (speech, images etc) would naturally generate multiple datasets with different neural network models for each one.  While these references are focused on training based on federated learning, the underlying applications are natural motivations for our setting as well. Multi-model learning has also been applied in real-time video surveillance applications [R5].
>
> We hope that the reviewer is convinced that our approach is practically important, novel and applicable to a broad range of applications. We have also included this explanation in Appendix I of the revised version of the paper.
>
> **References**
>
> R1. Bergstra, James, et al. "Algorithms for hyper-parameter optimization." Advances in neural information processing systems 24 (2011).
>
> R2. Zhou, Zhi-Hua. Ensemble methods: Foundations and Algorithms. CRC press, 2012.
>
> R3. Bhuyan, Neelkamal, Moharir, Sharayu, Joshi, Gauri, “Multi-Model Federated Learning with Provable Guarantees”, EAI Valuetools 2022
>
> R4. Da Silva, Rafael Valente, et al. "Multichannel ALOHA Optimization for Federated Learning With Multiple Models." IEEE Wireless Communications Letters 11.10 (2022): 2180-2184.
>
> R5. Wu, Yanzhao, Ling Liu, and Ramana Kompella. "Parallel Detection for Efficient Video Analytics at the Edge." 2021 IEEE Third International Conference on Cognitive Machine Intelligence (CogMI). IEEE, 2021.

---

> > ### Author Response · Authors · 2022-11-19
> > **Response to Reviewer 9BzY (part 2/2)**
> >
> > > “Notice that this replication scheme is also a candidate for the experimental results in Fig. 4, as s=15,n=256. I would like the authors to clarify if the above property is also true for the developed coding schemes. If we allowed some additional divisibility criteria to be satisfied among the parameters, then do simpler schemes such as replication match the performance of the developed schemes?”
> >
> > Thank you for bringing up this very interesting point. We have now added a section (Appendix G) to discuss this aspect. We will refer to the simplified GC scheme as GC-Rep. We note that the newly developed schemes can leverage the existence of the GC-Rep scheme (when parameters satisfy certain divisibility criteria). Specifically, in the case of M-SGC, the underlying GC can be replaced with GC-Rep if $\lambda+1$ divides $n$. Let this simplified scheme be called M-SGC-Rep. In the case of SR-SGC, simplification is possible if $(\lceil\frac{B\lambda}{W-1+B}\rceil+1)$ divides $n$. Let this scheme be called SR-SGC-Rep. The GC-Rep scheme offers the following advantages over GC (these advantages will be naturally inherited by SR-SGC-Rep and M-SGC-Rep schemes as well): (i) decoding becomes extremely simple (ii) there is improved straggler resiliency (the scheme can sometimes handle more than s stragglers).
> >
> > It is worth noting that the computational load for GC-based and GC-Rep-based schemes is the same. Although decoding in GC-Rep-based schemes is simpler, as decoding time may be included into the master's idle time (we discuss this in Appendix K), this will not have any impact on the total run time. However, enhanced straggler resilience of GC-Rep-based schemes may lessen the number of rounds in which the master must perform wait-outs, potentially reducing the overall run time.
> >
> > Under identical load conditions, the SR-SGC-Rep scheme always permits a strict superset of straggler patterns compared to the GC-Rep scheme, as is the case between SR-SGC and GC. Therefore, SR-SGC-Rep is anticipated to outperform GC-Rep. A direct comparison between M-SGC-Rep and GC-Rep is not feasible in terms of the straggler patterns tolerated by these schemes. However, because M-SGC-Rep requires a substantially lower computational load than GC-Rep, it is expected to perform better than the latter.
> >
> > > “There is no statement/result on whether the developed schemes can be improved, or gap to optimality.”
> >
> > We would like to bring to the reviewer’s attention that we do present some information theoretic bounds in Appendix F (also see Remark 3.4) which show that the gap between optimal load and achievable load of the M-SGC scheme decreases as $O(\frac{1}{W})$ for fixed $n$, $B$ and $\lambda$.
> >
> > Regarding the SR-SGC scheme, we note that it can tolerate straggler patterns conforming to s-stragglers-per-round model (with delay 0) or the bursty straggler model (with delay B) and has a load of $\frac{s+1}{n}$, which is far from optimal. From the converse results in Tandon et al. (2017), it is known that the load should be at least $\frac{s+1}{n}$ for any scheme which tolerates straggler patterns conforming to the $s$-stragglers-per-round model with delay-$0$. Thus, to approach optimality in terms of the bound, we might have to think of completely different schemes which cannot tolerate all straggler patterns conforming to the $s$-stragglers-per-round straggler pattern.
> >
> > > “In Example 3.3.1, the authors claim that "if the master did not receive g2i(t) in round t and round t+2,.... However, as per my understanding of the model, this can never happen for that example. If g2i(t) does not return, it is because the node straggles in iteration t. If it straggles in iteration t, it then returns that answer in round t+B=t+2, and as per the coding scheme, g2i(t) is necessarily repeated in round t+2. Please correct this or add clarifying explanations”
> >
> > We would like to clarify that the $\{n=4,B=2,W=3,\lambda=2\}$ M-SGC scheme that we discuss in Example 3.3.1 can tolerate any straggler pattern which conforms to either the (2,3,2)-bursty straggler model or else, the (2,4,2)-arbitrary straggler model (we note this in Prop. 3.2). The situation described is a valid scenario under the latter straggler model. i.e., it is possible for worker-$i$ to be a straggler in rounds t and t+2. However, as the reviewer pointed out, we understand that there is scope for confusion here and hence we have avoided mentioning this scenario in the revised manuscript.
> >
> > > “A small clarification in the explanation of Fig. 14: For a given normalized load, are there 80 "dots" on the y axis corresponding to each job? If this is the case, the figure is the result of several hundreds of jobs? If not, then it is unclear what the dots represent”
> >
> > We redrew this figure (Figure 17 in the new version)  to explicitly show the dependence of runtime on various choices of parameters.

---

### Official Review · Reviewer_82EB · 2022-10-31

**Confidence:** 4
**Correctness:** 4
**Technical Novelty And Significance:** 2
**Empirical Novelty And Significance:** Not applicable
**Recommendation:** 6

**Clarity, Quality, Novelty And Reproducibility:**

The paper is clear overall and the proposed scheme is novel within the context of coded distributed gradient descent. I have the following three comments/suggestions on the writing:

a. The audience, especially at ICLR, may not be familiar with the use of erasure codes for straggler mitigation in distributed computing. Therefore, I feel it would be better to provide some background on the general area of coded computing to make the paper more accessible at this venue.

b. Likewise it may be beneficial to provide some details on the Gilbert Elliot model and its deterministic counterpart to clarify why it is a suitable model for stragglers.

c. Please include a conclusion section.

**Strength And Weaknesses:**

Strengths:

1. The approach considers sequential computations and is designed based on the trends in straggling over time. The temporal aspect of straggling has not been considered by most prior coded distributed computing works and is arguably more practical in real scenarios (as also demonstrated by the experiments in this work).

2. The proposed approach is faster than baselines in real serverless computing settings (AWS Lambda) and reduces the computation load at workers which may be especially relevant for emerging frameworks like serverless computing where the individual instances do not have a lot of computing capacity.

Weakness:

1. The approach will improve over prior gradient coding works only when a finite amount of delay ($T>0$) can be tolerated. The authors consider a scenario where multiple neural networks are trained on the same dataset in their experiments where such a delay can be tolerated. However, it is unclear when this would happen in practice because typically gradient descent is a sequential operation with successive iterations of a single model being performed where such a delay would not be acceptable because the gradient from each round is needed for the next round. Please provide some examples/references to back this claim.

2. The proposed approach seems to strongly depend on the choice of hyperparameters ($B$, $W$, etc.). Indeed, the authors perform additional experiments to select the right hyperparameters. Given, the extra cost/latency of performing these added experiments, and the extra cost/latency of decoding the coded gradients, neither of which is included in the time measurements, it is unclear if the proposed schemes will give any net improvements over the uncoded scheme, especially since model training is a one-time operation where a small amount of straggling may actually not be a very significant issue. Therefore I would like to see either that the gain is significant after taking these factors into consideration or that incorrect/different choices of hyperparameters for the same setting do not lead to much differences in performance since that would imply that the extra experiments to choose the hyperparameters will not be needed.

**Summary Of The Paper:**

The paper proposes a new coded distributed gradient descent approach to mitigate the effect of stragglers in the distributed computation of gradients. The authors combine prior schemes on coded distributed gradient computation with repetition of unfinished tasks from straggling nodes across computation rounds with the assumption being that the overall computation can tolerate a finite amount of delay. Experiments on a distributed gradient descent task on AWS Lambda shows improvements over prior uncoded and coded approaches.

**Summary Of The Review:**

The paper introduces a new coded distributed gradient computation approach for mitigating stragglers in distributed gradient descent. While the approach improves over baselines if a finite delay in the computation can be tolerated, it is unclear where such delays would be acceptable in practice. Moreover, given the added cost/latency associated with choosing the hyperparameters for this approach the net gain over the uncoded approach is not yet clear.

---

> ### Author Response · Authors · 2022-11-19
> **Response to Reviewer 82EB (part 1/3)**
>
> We thank the reviewer for their careful reading and constructive feedback towards improving our manuscript. We address the reviewer’s points below and have updated our manuscript accordingly.
>
> > “It is unclear when this would happen in practice because typically gradient descent is a sequential operation… where such a delay would not be acceptable.”
>
> First, we would like to clarify why it is not necessary that the gradient computation of each round be immediately available in the next round. We consider training $M=4$ neural networks, denoted by $NN_1$, $NN_2$, $NN_3$, and $NN_4$ (as we did in the experimental section). For neural network $NN_j$ (where $j=1,2,3,4$), we let ${\bf w}_j^{(0)}$ denote the initial weights and ${\bf w}_j^{(i)}$ denote the weights after a total of $i$ rounds of gradient descent updates i.e.,
>
> ${\bf w}_j^{(i)} = {\bf w}_j^{(i-1)} - \epsilon_j^{(i)} {\bf g}_j^{(i-1)}$
>
> where ${\bf g}_j^{(i-1)}$ denotes the gradient associated with neural network $NN_j$ with weights $\bf w_j^{(i-1)}$ and $\epsilon_j^{(i)}$ denotes the learning rate.
>
> We assume that the training of $NN_1$, $NN_2$, $NN_3$, and $NN_4$ is interleaved across rounds. In particular, the model updates for the interleaved training uses the following schedule:
> - Weights of $NN_1$ are updated in the following rounds: 1 (Initialization), 5, 9, 13, ...
> - Weights of $NN_2$ are updated in the following rounds: 2 (Initialization), 6, 10, 14,…
> - Weights of $NN_3$ are updated in the following rounds: 3 (Initialization), 7, 11, 15,…
> - Weights of $NN_4$ are updated in the following rounds: 4 (Initialization), 8, 12, 16,…
>
> Thus if we consider the training of one such neural network, say $NN_1$, then there are the following steps:
>
> - **Step 1**: At the start of round $4i+1$, the server will generate ${\bf w}_1^{(i)}$. It will have decoded the gradient vector ${\bf g}_1^{(i-1)}$ by the end of round $4i$ and performed the SGD update noted above.
> - **Step 2**: The server will issue a request at the start of round $4i +1$ to the client nodes to compute the associated partial gradients such that the gradient vector $\bf{g}_1^{(i)}$ can be computed by the end of round $4i+4$.
>
> By following a similar approach for each model it is clear that for neural network $NN_j$, when computing the gradient ${\bf g}_j^{(i)}$, a request will be issued by the server to the clients at the start of round $4i+j$ and the computation must be completed by the end of round $4i+j +3$. Thus our interleaved approach is designed so that each of the gradient vectors can be decoded within a span of $M=4$ rounds.
>
> Our method can be viewed as a pipelined approach for training of multiple neural networks. Note that in our current setting we assume that the parameters of only one neural network model can be updated in each round, which arises in resource constrained devices. Our method also naturally complements other approaches for parallel training of multiple neural network models, and leverages on the temporal structure of straggler patterns to achieve speedups. We also emphasize that we do not require that the multiple models we consider be trained on the same dataset. Each model can be trained on a different dataset. We also do not require that the architecture of the neural network models be identical. Nevertheless, we point out that our setting would be most efficient when the compute time for the gradients for each model is approximately the same. We believe this is a rather benign requirement. Finally, we discuss a few applications where multiple-model training arises naturally.
>
> 1. In the training of deep learning models, we are often required to perform a search over various hyperparameters and this is done through some form of a grid search [R1]. Each choice of hyperparameter corresponds to a new model. Ultimately the ideal hyperparameters are selected through a validation set.
> 2. In ensemble learning [R2], a number of models need to be trained simultaneously. Their predictions are then combined through some averaging mechanism.
> 3. Since our approach can use completely different datasets for each of the models, it is also applicable in settings of “multi-model learning” [R3, R4], where multiple datasets are used for different models. For instance, sensors deployed in time-varying or periodic environment (e.g. day/night camera images, orbiting satellite data etc) or collecting different modalities (speech, images etc) would naturally generate multiple datasets with different models for each one.  While these works are focused on training based on federated learning, the underlying applications are natural motivations for our setting as well. Multi-model learning has also been applied in real-time video surveillance applications [R5].
>
> We hope that the reviewer is convinced that our approach is practically important, novel and applicable to a broad range of applications. We have also included this explanation in Appendix I of the revised version of the paper.

---

> > ### Author Response · Authors · 2022-11-19
> > **Response to Reviewer 82EB (part 2/3)**
> >
> > > “Given, the extra cost/latency of performing these added experiments, and the extra cost/latency of decoding the coded gradients, neither of which is included in the time measurements, it is unclear if the proposed schemes will give any net improvements over the uncoded scheme”
> >
> > To address the reviewer's point on overheads, we would like to explain how both sources of possible overheads, namely the parameter selection time and decoding time can be eliminated. For the former, in practice, the master node can start with uncoded training and switch to coded training after a certain number of rounds. The coding parameters are selected using the response times from the first uncoded section. This idea is experimentally tested in Appendix K.2 (please see Figure 18) to demonstrate that even after taking the parameter selection time into account, our coding schemes still outperform uncoded training and baseline GC.
> >
> > As for the decoding time, we would like to argue that whenever $M > T+1$  models are trained concurrently, the master node can use its idle time to perform decoding to avoid any holdups. This is indeed the case for all coding schemes used in the experiments ($T < M-1 = 3$). As an example, let us consider the SR-SGC$(B=2, W=3, \lambda=23)$ scheme in Section 4.2, where $M=4$ and $T=2$. Calculations of gradients for model 1 start at rounds $t_1=1, t_2=5, t_3=9, \cdots$. Calculating the first gradients of model 1 starts at round $t_1=1$. By the end of round $t_1+T=3$, the gradients are ready to be decoded. Note that the second gradients of model 1 are to be calculated at round $t_2=5$. Therefore, the master node can perform the decoding during its idle time over round $4$, when it is waiting for the worker nodes (note that as stated in Table 4, the longest decoding time is shorter than the fastest round time). This way, the gradients are decoded before round $t_2=5$ begins and no decoding overhead is imposed. We added Appendix K for a detailed discussion on overhead analysis.
> >
> > Regarding sensitivity to hyperparameters, we added a comprehensive discussion in Appendix J.1. We show that the performance of our proposed schemes is fairly insensitive to the choice of B and W as long as W and B are close. Moreover, for M-SGC the choice of $\lambda$ is not critical as long as it is above a certain threshold, but for SR-SGC it is an important consideration as it affects the load heavily.
> >
> > We also evaluate the sensitivity of parameter selection to the number of rounds used in the selection process. Table 3 lists the selected parameters using different numbers of rounds. For each unique set of parameters, we performed the experiments discussed in Section 4.2 ten times and included the average runtime. As expected, we observe a general trend of improvement in total training time by increasing rounds used. Moreover, for M-SGC, even a few rounds are enough to tune the coding scheme, as M-SGC with parameters selected over only 10 rounds outperforms other coding schemes where as high as 80 rounds are used.
> >
> > In conclusion, we can answer that the gain is indeed significant after taking overhead times into consideration. Moreover, choosing hyperparameters using only a few rounds does not degrade the performance of M-SGC. General recommendations for selecting the hyperparameters are also included in Remark J.1 to guide easier hyperparameter selection.

---

> > > ### Author Response · Authors · 2022-11-19
> > > **Response to Reviewer 82EB (part 3/3)**
> > >
> > > > “It would be better to provide some background on the general area of coded computing… Provide some details on the Gilbert Elliot model and its deterministic counterpart…”
> > >
> > > We have added a new section (Appendix B) which provides a brief overview on the relevance of erasure coding for straggler mitigation in distributed computing. We discuss two toy examples that demonstrate the fundamental concepts; the first example focuses on the pioneering work by Lee et al. [R6] on distributed matrix multiplication, and the next one is on distributed SGD (gradient coding) by Tandon et al. [R7].
> > >
> > > We have also added a new section (Appendix C) providing details on the Gilbert-Elliot (GE) model. The GE model suggests that straggling behavior occurs periodically and is often followed by non-straggling behavior. In distributed systems, stragglers occur due to reasons such as resource sharing, communication bottlenecks, routine maintenance activities, etc. The periodic spikes in latency that are caused by several of these factors (see [R8], [R9]) are naturally captured by the GE model. The authors of [R10] make an empirical observation that the GE model can accurately track the state transitions of workers on an Amazon EC2 cluster, particularly in the context of applying erasure codes for straggler mitigation. In the current study, we approximate the GE model using deterministic, sliding-window-based models since they are more tractable in terms of code design. We provide intuitive justification for why sliding-window-based models make a good approximation for the GE model in the appendix. As our experiments indicate, such a design strategy finally leads to techniques that outperform the baseline GC on an AWS Lambda cluster.
> > >
> > > > “conclusion section”
> > >
> > > We have added a conclusion section in the revised manuscript.
> > >
> > >
> > > **References**
> > >
> > > R1. Bergstra, James, et al. "Algorithms for hyper-parameter optimization." Advances in neural information processing systems 24 (2011).
> > >
> > > R2. Zhou, Zhi-Hua. Ensemble methods: Foundations and Algorithms. CRC press, 2012.
> > >
> > > R3. Bhuyan, Neelkamal, Moharir, Sharayu, Joshi, Gauri, “Multi-Model Federated Learning with Provable Guarantees”, EAI Valuetools 2022
> > >
> > > R4. Da Silva, Rafael Valente, et al. "Multichannel ALOHA Optimization for Federated Learning With Multiple Models." IEEE Wireless Communications Letters 11.10 (2022): 2180-2184.
> > >
> > > R5. Wu, Yanzhao, Ling Liu, and Ramana Kompella. "Parallel Detection for Efficient Video Analytics at the Edge." 2021 IEEE Third International Conference on Cognitive Machine Intelligence (CogMI). IEEE, 2021.
> > >
> > > R6. Lee, Kangwook, et al. "Speeding up distributed machine learning using codes." IEEE Transactions on Information Theory 64.3 (2018): 1514-1529.
> > >
> > > R7. Tandon, Rashish, et al. "Gradient coding: Avoiding stragglers in distributed learning." International Conference on Machine Learning. PMLR, 2017.
> > >
> > > R8. Dean, Jeffrey, and Luiz André Barroso. "The tail at scale." Communications of the ACM 56.2 (2013): 74-80.
> > >
> > > R9. Bolot, Jean-Chrysotome. "End-to-end packet delay and loss behavior in the Internet." ACM SIGCOMM. 1993.
> > >
> > > R10. Yang, Chien-Sheng, Ramtin Pedarsani, and A. Salman Avestimehr. "Timely coded computing." 2019 IEEE International Symposium on Information Theory (ISIT). IEEE, 2019.

---

### Author Response · Authors · 2022-11-19
**Dear Reviewers**

We would like to express our gratitude to all the reviewers for their inputs. We have made significant additions (highlighted in blue color) to the manuscript to take into account all the comments from the reviewers. Among other revisions/additions to the paper, we have included the following in particular;

- Discussion on overheads due to decoding & parameter selection and how we ensure that these overheads will not have any impact on the performance of our coding schemes.
- Additional experimental results based on training ResNet-18 models on CIFAR-100.
- Discussion on the applicability of our schemes to various practical use cases.
- Discussion on sensitivity to parameters.

---

### Decision · Program_Chairs · 2023-01-20

**Decision:**

Accept: poster

**Justification For Why Not Higher Score:**

The paper although solid is quite limited in it's scope as it addresses a problem in the intersection of ICLR, ML, Sys, Coding theory communities.

**Justification For Why Not Lower Score:**

The paper presents a concrete set of advances both theoretical and practical in the area of codes for machine learning.

**Metareview: Summary, Strengths And Weaknesses:**

In this paper, the authors introduce two new techniques for distributed gradient computation in the presence of stragglers. The techniques combine Gradient Coding (GC) with selective repetition of tasks and adaptive multiplexing across workers and rounds. The theoretical analysis and experiments on an AWS Lambda cluster showed improved performance over the baseline GC scheme, leading to a 16% reduction in runtime.

The reviewers and myself identified several strengths, and only a few weaknesses about this work.

Strengths:
- Proposed algorithm is novel: Combines prior schemes on coded distributed gradient and provides new extensions
- Thorough experimental evaluation, in which proposed methods are faster than baselines in real serverless settings (e.g., AWS Lambda)
- Reduces the computation load at workers
- Theoretical analysis provided for computational load
- Well written with enough details for reproducing experiments
- The schemes cover various straggler models (e.g., bursty, arbitrary, fixed number per round)

Weaknesses:
- Unclear where finite delay in computation can be tolerated in practice
- The schemes depend strongly on several hyperparameters (B, W, etc.), and there's added cost in choosing them
- No convergence proof provided
- Presentation is often hard to follow due to the heavy and involved notation

The authors incorporated several of the reviewer's comments and made several revisions / additions to their paper to address concerns. These included a discussion on overheads due to decoding and parameter selection, additional experimental results, a discussion on applicability to various practical use cases and on sensitivity to hyperparameter selection error.

Overall, some concerns still remain, but this is a useful paper that will be of interest to a small subset of ML+Sys community within ICLR.

**Note From Pc:**

if the above contains the word "oral" or "spotlight" please see: "oral" presentation means -> notable-top-5% and "spotlight" means -> notable-top-25%. As stated in our emails, we are disassociating presentation type from AC recommendations

**Summary Of Ac-Reviewer Meeting:**

N/A